# LEARNING REPRESENTATIONS FROM TEMPORALLY SMOOTH DATA

## ABSTRACT

Events in the real world are correlated across nearby points in time, and we must learn from this temporally "smooth" data. However, when neural networks are trained to categorize or reconstruct single items, the common practice is to randomize the order of training items. What are the effects of temporally smooth training data on the efficiency of learning? We first tested the effects of smoothness in training data on incremental learning in feedforward nets and found that smoother data slowed learning. Moreover, sampling so as to minimize temporal smoothness produced more efficient learning than sampling randomly. If smoothness generally impairs incremental learning, then how can networks be modified to benefit from smoothness in the training data? We hypothesized that two simple brain-inspired mechanisms – leaky memory in activation units and memory-gating – could enable networks to rapidly extract useful representations from smooth data. Across all levels of data smoothness, these brain-inspired architectures achieved more efficient category learning than feedforward networks. This advantage persisted, even when leaky memory networks with gating were trained on smooth data and tested on randomly-ordered data. Finally, we investigated how these brain-inspired mechanisms altered the internal representations learned by the networks. We found that networks with multi-scale leaky memory and memory-gating could learn internal representations that "un-mixed" data sources which vary on fast and slow timescales across training samples. Altogether, we identified simple mechanisms enabling neural networks to learn more quickly from temporally smooth data, and to generate internal representations that separate timescales in the training signal.

## 1 INTRODUCTION

Events in the world are correlated in time: the information that we receive at one moment is usually similar to the information that we receive at the next. For example, when having a conversation with someone, we see multiple samples of the same face from different angles over the course of several seconds. However, when we train neural networks for categorization or reconstruction tasks, we commonly ignore temporal ordering of samples and use randomly ordered data. Given that humans can learn robustly and efficiently when learning incrementally from sequentially correlated, it is important to examine what kinds of architectures and inductive biases may support such learning (Hadsell et al., 2020). Therefore, we asked how does the sequential correlation structure in the data affect learning in neural networks that are performing categorization or reconstruction of one input at a time? Moreover, we asked: which mechanisms can a network employ to exploit the temporal autocorrelation ("smoothness") of data, without needing to perform backpropagation through time (BPTT) (Sutskever, 2013)?

We investigated this question in three stages. In the first stage, we examined the effects of temporally smooth training data on feedforward neural networks performing category learning. Here we confirmed that autocorrelation in training data slows learning in feeforward nets.

In the second stage, we investigated conditions under which these classifier networks might take advantage of smooth data. We hypothesized that human brains may possess mechanisms (or inductive biases) that maximize the benefits of learning from temporally smooth data. We therefore tested two network mechanisms inspired by properties of cortical circuits: leaky memory (associated with

autocorrelated brain dynamics), and memory gating (associated with rapid changes of brain states at event boundaries). We compared the performance of these mechanisms relative to memoryless networks and also against a long short-term memory (LSTM) architecture trained using BPTT.

Finally, having demonstrated that leaky memory can speed learning from temporally smooth data, we studied the internal representations learned by these neural networks. In particular, we showed that networks with multi-scale leaky memory and resetting could learn internal representations that separate fast-changing and slow-changing data sources.

## 2 Related Work

**Effects of sampling strategies on incremental learning.** The ordering of training examples affects the speed and quality of learning. For example, learning can be sped by presenting "easier" examples earlier, and then gradually increasing difficulty (Elman, 1993; Bengio et al., 2009; Kumar et al., 2010; Lee & Grauman, 2011). Similarly, learning can be more efficient if training data is organized so that the magnitude of weight updates increases over training samples (Gao & Jojic, 2016).

Here, we do not manipulate the order based on item difficulty or proximity to category boundaries; we only explore the effects of ordering similar items nearby in time. We aim to identify mechanisms that can aid efficient learning across many levels of temporal autocorrelation, adapting to what is present in the data. This ability to adapt to the properties of the data is important in real-world settings, where a learner may lack control over the training order, or prior knowledge of item difficulty is unavailable.

**Potential costs and benefits of training with smooth data.** In machine learning research, it is often assumed that the training samples are independent and identically distributed (iid) (Dundar et al., 2007). When training with random sampling, one can approximately satisfy iid assumptions because shuffling samples eliminates any sequential correlations. However, in many real-world situations, the iid assumption is violated and consecutive training samples are strongly correlated.

Temporally correlated data may slow learning in feedforward neural networks. If consecutive items are similar, then the gradients induced by them will be related, especially early in training. If we consider the average of the gradients induced by the whole training set as the "ideal" gradient, then subsets of similar samples provide a higher-variance (i.e. noisier) estimate of this ideal. Moreover, smoothness in data may slow learning due to *catastrophic forgetting* (French, 1999). Suppose that, for smoother training, we sample multiple times from a category before moving to another category. This means that the next presentation of each category will be, on average, farther apart from its previous appearance. This increased distance could lead to greater forgetting for that category, thus slowing learning overall. On the other hand, smoother training data might also benefit learning. For example, there may be some category-diagnostic features that will not reliably be extracted by a learning algorithm unless multiple weight updates occur for that feature nearby in time; smoother training data would be more liable to present such features nearby in time.

## 3 Research Questions and Hypotheses

1. How does training with temporally smooth data affect learning in feedforward networks? *In light of the work reviewed above, we hypothesized that temporally smooth data would slow learning in feedforward nets.*

2. How can neural networks benefit from temporally smooth data, in terms of either learning efficiency or learning more meaningfully structured representations? *We hypothesized that a combination of two brain-inspired mechanisms — leaky memory and memory-resetting — could enable networks to learn more efficiently from temporally smooth data, even without BPTT.*

## 4 Effects of temporal smoothness in training data on learning in feedforward neural networks

We first explored how smoothness of data affects the speed and accuracy of category learning (classification) in feedforward networks. See Appendix A.1 for similar results with unsupervised learning.

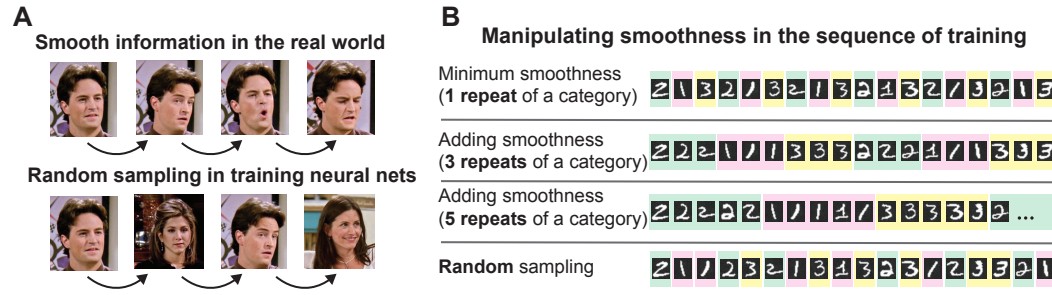

Figure 1: Temporal smoothness in the real world and in neural network training. *A) Top: smooth information in the real world. Bottom: randomly ordered data in training neural networks.[2] B) Manipulating smoothness levels in training data using the ordering of training samples. Colored rectangles indicate the amount of smoothness induced by repeating a category.*

## 4.1 METHODS

### 4.1.1 MANIPULATING SMOOTHNESS IN TRAINING DATA

We manipulated smoothness in training data by varying the number of consecutive samples drawn from the same category. We began each training session by generating a random "category order", which was a permutation of the numbers from 1 to $N$ (e.g. the ordering in Figure 1.B is 2-1-3). The same category order was used for all smoothness conditions in that training session.

To sample with minimum smoothness, we sampled exactly one exemplar from each category, before sampling from the next category in the category order (1 repeat) (Figure 1.B). This condition is called "minimum smoothness" because all consecutive items were from different categories, and there were not more examples from a category until all other categories were sampled. We increased smoothness by increasing the number of consecutive samples drawn from each category (3 repeats and 5 repeats in Figure 1.B). Finally, we also used the standard random sampling method, in which items were sampled at random, without replacement, from the training set (Figure 1.B). The training set was identical across all conditions, as was the order in which samples were drawn from within a category (Figure 1.B).

### 4.1.2 FEEDFORWARD NEURAL NETWORK

**Dataset**. We tested MNIST, Fashion-MNIST, and synthetic datasets containing low category overlap (LeCun et al., 2010; Xiao et al., 2017). An example synthetic dataset is shown in Appendix A.2. For creating synthetic datasets, we used Numpy (Harris et al., 2020). For creating and testing the models, we used PyTorch(Paszke et al., 2019).

**Learning rule and objective function.** We used backpropagation with both mean squared error (MSE) and cross-entropy (CE) loss functions. The results reported here are using MSE, primarily for the ease of comparison with later reconstruction error measures in this manuscript. However, the same pattern was observed using CE loss, as shown in Appendix A.3. Also, it has been shown MSE loss provides comparable performance to commonly utilized classification models with CE loss function (Illing et al., 2019). To test incremental learning, we employed stochastic gradient descent (SGD), updating weights for each training sample.

**Optimization, initialization, and activation function.** We tested the model both with and without RMSprop optimization, along with Xavier initialization (Tieleman & Hinton, 2012; Glorot & Bengio, 2010). We applied ReLU to hidden units and Softmax or Sigmoid to the output units.

**Hyperparameters.** For MNIST and Fashion-MNIST, we used a 3-layer fully connected network with (784, 392, 10) dimensions and a learning rate of 0.01. The learning rate was not tuned for a specific condition. We used the same learning rate across all conditions; only smoothness varied across conditions. To compensate for potential advantage of a specific set of hyperparameters for a specific condition, we ran 5 runs, each with a different random weight initialization, and reported

---

[2]Photos in this section are taken from the FRIENDS TV series, Warner Brothers (Kauffman et al., 1994).

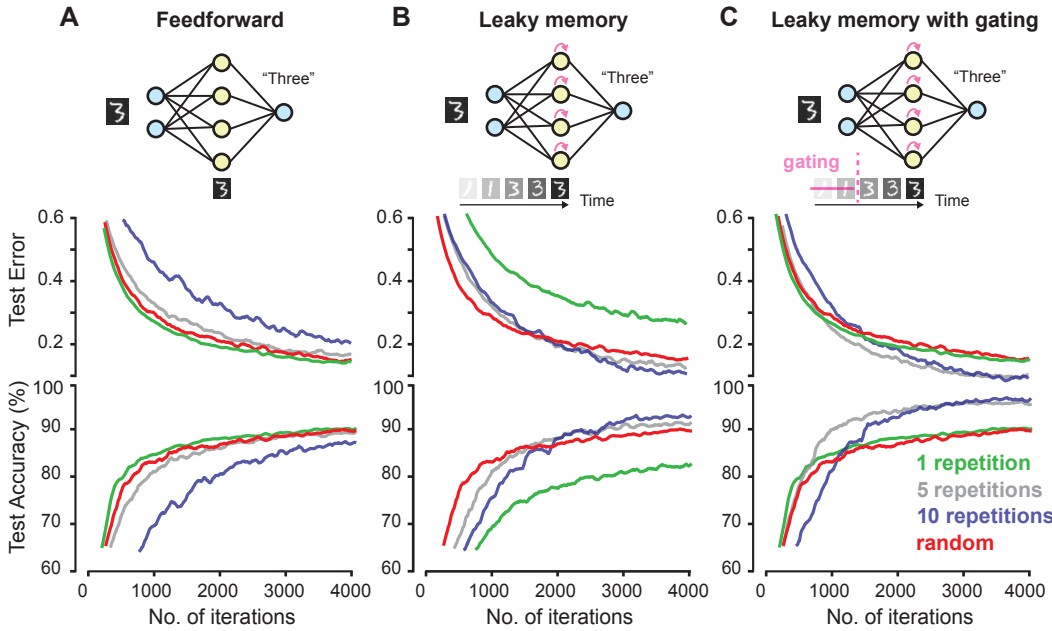

Figure 2: Neural architectures for classifying temporally smooth data. *A) Test error (MSE loss) and test accuracy in SGD training of a feedforward neural network (MNIST data) across different smoothness levels. B) The same as A, for a neural network with leaky memory in internal representations. C) The same as A, for a neural network with leaky memory and gating mechanism. Random sampling in all 3 plots are identical and can be used as a common reference. [Curves in this figure have been averaged over 5 runs with different initialization and were further smoothed using a 100-iteration moving average.]*

the averaged results. For hyperparameters in synthetic dataset see Appendix A.2. When RMSprop was implemented, $\beta_1$ and $\beta_2$ were set to 0.9 and 0.99, respectively (Ruder, 2016).

## 4.2 RESULTS

Smooth training data slowed incremental learning (Figure 2.A). Moreover, minimum smoothness yielded more efficient learning than random sampling (Figure 2.A). These observations generalized across all tested datasets and across MSE and CE loss, with and without RMSprop optimization.

## 4.3 DISCUSSION

The superiority of minimum smoothness over other conditions suggests that any level of smoothness slows incremental learning, even the smoothness that can occur by chance in random sampling (Figure 2.A). Therefore, given a fixed time budget for training, a sampling strategy that minimizes smoothness can reach a higher performance than random sampling.

Sampling with minimum smoothness may be advantageous because it reduces the representation overlap across consecutive training items. Catastrophic forgetting can be reduced by decreasing the overlap between learned representations, for example, via orthogonalization (French, 1999). Though we did not explicitly seek to reduce interference by sampling with minimum smoothness, this method does likely reduce the representational overlap of nearby items. In addition, training with minimum smoothness may improve learning by maintaining a near-uniform distribution of sampled categories. Training with "low-discrepancy" sequences, such as those with uniformly distributed data, avoids classification bias and enhances learning (Iwata & Ishii, 2002; Mishra & Rusch, 2020).

## 5 EXPLOITING TEMPORAL SMOOTHNESS IN TRAINING DATA FOR LEARNING IN NEURAL NETWORKS

Although temporally-correlated data slows learning in feedforward nets, it appears that humans are able to rapidly extract meaningful representations in such settings, even while learning incrementally. How might our brains maximize the benefits of temporally smooth training sets? Two properties of cortical population dynamics appear especially relevant to incremental learning: (i) all cortical dynamics exhibit *autocorrelation* on the scale of milliseconds to seconds, so that correlation in consecutive internal states is unavoidable (Murray et al., 2014; Honey et al., 2012; Bright et al., 2020); (ii) neural circuits appear to shift state suddenly at *event boundaries*, and this appears to be associated with "resetting" of context representations (DuBrow et al., 2017; Chien & Honey, 2020; Baldassano et al., 2018). We hypothesized that these two neural properties represent an *inductive bias* in cortical learning. In particular, we hypothesized that (i) data sampled from a slowly-changing environment may contain important features that are stable over time, which can be better extracted by mixing current input with a memory of recent input; and (ii) the interference of irrelevant prior information can be reduced by "resetting" memory at boundaries between events. Therefore, we examined how neural network learning was affected by two brain-inspired mechanisms: (i) leaky memory in internal representations; (ii) a memory gating mechanism that resets internal representation at transitions between categories.

### 5.1 BRAIN-INSPIRED NEURAL ARCHITECTURE FOR SUPERVISED LEARNING

Can brain-inspired architectural tweaks – leaky memory and memory gating – increase the efficiency of learning in supervised classification tasks?

#### 5.1.1 METHODS

**Leaky memory:** We added leaky memory to the internal representations (hidden units) by linearly mixing them across consecutive time points. Hidden unit activations were updated according to following function:

$$H(n) = \alpha H(n-1) + (1 - \alpha)\text{ReLU}(W_{IH}I(n)) \tag{1}$$

where $H(n)$ is the state of the hidden units for trial $n$, $I(n)$ is the state of the input units for trial $n$, $\alpha$ is a leak parameter, $W_{IH}$ are the connections from the input layer to the hidden layer, and ReLU is a rectified linear activation. We set $\alpha = 0.5$ in these experiments.

**Memory Gating:** In order to reduce the interference between items from different categories in the leaky memory, we employed a gating mechanism to reset memory at the transitions between categories. Therefore, if sample n was drawn from a category other than the category of sample $n - 1$, then we set $\alpha = 0$ in Eq.(1) on that trial n (Figure 2.C).

For the learning rule, we used backpropagation, however the gradient computation did not account for the fact that the neurons were leaky. Therefore, the update rule in [leaky memory + reset] model is different from the common update rule in recurrent models (e.g. LSTM). LSTM uses backpropagation through time (BPTT), which is implausible for biological settings. In learning with BPTT, the same neurons must store and retrieve their entire activation history (Sutskever, 2013; Lillicrap & Santoro, 2019). In contrast, in the [leaky memory + reset] model, neurons only use local information from their most recent history. Therefore, it is computationally much simpler because it does not require to maintain the whole history and to compute the gradient relative to all that history.

Optimization and initialization methods, and the hyperparameters were identical to those used in training and testing feedforward neural networks.

#### 5.1.2 RESULTS

Smoothness in training data increased learning efficiency in learners with leaky memory, as shown in Figure 2.B. This result is in contrast to the detrimental effects of smoothness in memoryless learners (Figure 2.A). Moreover, adding a gating mechanism to the leaky memory units further increased their learning (Figure 2.C). In learners with leaky memory and gating, all levels of smoothness significantly outperformed random sampling and sampling with minimum smoothness (1 repeat)

(Figure 2.C). These findings generalized across MNIST, Fashion-MNIST, and synthetic datasets [Appendix A.4].

### 5.1.3 DISCUSSION

When data sampled at a given moment shares category-relevant features with recent samples, learners with leaky memory were able to exploit this property for more efficient category learning (Figure 2.B, C). Importantly, the resetting mechanism prevented the mixing of hidden representations from samples of different categories, allowing the system to benefit most from the data smoothness, while not suffering from between-category interference.

**Why does averaging of current and prior states produce more efficient learning from sequentially correlated data streams?** Our working hypothesis is that averaging across multiple members of the same category increases (in some datasets) the proportion of variance in the hidden units that is associated with category-diagnostic features. This hypothesis predicts that if consecutive items in the data stream, do not share any local features, then the benefits of leaky memory will be eliminated. We confirmed this prediction empirically (Appendix A.7).

Importantly, networks with leaky memory and resetting surpassed the performance of feedforward networks, for all levels of smoothness (Figure 2.A, C). Also, leaky memory networks trained with smooth data surpassed feedforward networks, even when tested on data streams that were not smooth. This finding is notable because it indicates that the leaky memory networks learned better single-exemplar representations, because they could generalize to novel temporal contexts. Moreover, the superior learning was obtained without BPTT, only using a linear mixture of activations over time-steps, which is easy to implement in brain dynamics (Honey et al., 2012; Murray et al., 2014) .

**How does the [leaky memory + reset] net compare with a more flexible recurrent net trained with BPTT?** The leaky-memory model with reset is trained without BPTT, but it is important to compare this to the performance of a more flexible model that can directly learn from task-relevant temporal structure. We found that an LSTM trained with BPTT was able to benefit from training with smooth data, learning more slowly at first, but ultimately achieving the lowest test error of all models (See Appendix A.8 and A.9). However, the performance advantage of the LSTM trained with BPTT was not preserved when the models were tested out-of-domain. In particular, when models were trained on data that contained temporal smoothness, but tested on data with minimum smoothness (1 repetition per category), the leaky-memory with reset model showed the best performance of all models (See Appendix A.10). We interpret these results as evidence that the LSTM has a much more flexible architecture, and via BPTT it can be calibrated to the exact structure of the training data stream (e.g. there are precisely 5 repetitions in a block). Conversely, the leaky memory model with reset is more biologically plausible, it is trained without BPTT, it showed performance competitive with the LSTM in this setting, and it generalized better across different levels of temporal smoothness. Note that we found these results despite the fact that in the LSTM, the gradient updates are mathematically optimized for the task (via BPTT), whereas, in the [leaky memory + reset] model, the gradient computations do not account for the recurrence in the network at all.

**Are the benefits of leaky-memory due to a form of gradient averaging, analogous to mini-batching?** Leaky-memory networks average activations over time, while batching averages gradients over time. The two mechanisms appear to differ, because leaky-memory effects can be reversed when the training categories contain non-overlapping features (Appendix A.7) and leaky-memory and mini-batching affect performance in different ways as a function of the amount of category repetition (Appendix A.5 and A.6).

### 5.2 BRAIN-INSPIRED ARCHITECTURES FOR UNSUPERVISED LEARNING ACROSS TEMPORAL SCALES

In the real world, we may need to learn from data with multiple levels of smoothness. For instance, returning to the example of having a face-to-face conversation: the features around a person's mouth change quickly, while their face's outline changes more slowly (Figure 3.A). Moreover, there are no pre-defined labels to support the learning of representations in this setting. We hypothesized that neural networks equipped with multi-scale (i.e. fast and slow) leaky memory could learn more

meaningful representations in this settings, by separating their representations of structures that vary on fast and slow timescales.

### 5.2.1 METHODS

**Dataset.** To test the un-mixing abilities of our networks, we synthesized simplified training datasets which contained three levels of temporal structure. The input to the model at each time point consisted of 3 subcomponents (top, middle, bottom), and each subcomponent had two elements. Each subcomponent was generated to express a different level of smoothness over time: for example, the top, middle and bottom rows changed feature-category every 1, 3 or 5, iterations, respectively (Figure 3.B). The individual features sampled at each time were generated as the sum of (i) an underlying binary state variable (which would switch every 1, 3 or 5 iterations) and (ii) uniformly-distributed noise (Appendix A.12). As a result, the model was provided with features that varied at 3 timescales: fast (top row), medium (middle row), and slow (bottom row). For creating the dataset, and designing and analyzing the models we used Numpy (Harris et al., 2020).

**Architectures.** We used the same brain-inspired mechanisms for unsupervised learning models: leaky memory and gating mechanisms. To evaluate the effectiveness of the added mechanisms, we compared 5 types of autoencoder (AE) models (See Figure 3.C): i) Feedforward AE; ii) AE with leakymemory in internal representations; iii) AE with multi-scale leaky memory in internal representations, inspired by evidence showing that levels of processing in the brain can integrate information at different time-scales (Honey et al., 2012; Murray et al., 2014; Bright et al., 2020), and that multiple time-scales are present even within a single circuit (Bernacchia et al., 2011; Ulanovsky et al., 2004); iv) AE with leaky memory in internal representations and boundary-sensitive gating, motivated by the evidence showing that processing in cortical circuits are sensitive to event-boundaries and these boundaries can shift learned representations (DuBrow et al., 2017; Chien & Honey, 2020); and (v) AE with multi-scale leaky memory in internal representations and boundary-sensitive gating. Gating mechanism was sensitive to change in the input stream. It would use information from current and previous input to decide to reset memory when the change passed a threshold (see Appendix A.11) (Chien & Honey, 2020).

**Learning algorithm, optimization, and initialization.** We used backpropagation with MSE loss, both with and without RMSprop optimization method, and Xavier initialization (Tieleman & Hinton, 2012; Glorot & Bengio, 2010). We applied ReLU and Sigmoid as activation functions for hidden and output units, respectively.

**Hyperparameters.** To implement leaky memory at multiple scales, we varied the time constants across the nodes in the hidden layer. Thus, the variable in Eq.(1) was set to 0, 0.3, and 0.6 for "short memory", "medium-memory", and "long-memory" nodes, respectively. The networks were 3-layer, fully connected autoencoders with (6, 3, 6) dimension. Learning rate was 0.01. In cases where RMSprop was implemented, the beta-1 and beta-2 were set to 0.9 and 0.99. For leaky memory in internal representations in Eq.(1) was set to 0.5 (See Figure 3.C).

**Un-mixing Measures.** We measured the network's ability to "un-mix" the time-scales of its input. By un-mixing, we mean learning representations that selectively track distinct latent sources that generated features within each training sample. In particular, we tested whether no-memory, short-memory, and long-memory nodes in the network would track the fast-, medium-, and slow-changing features in the data. To this end we measured the Pearson correlation between each hidden unit (no-memory, short-memory, and long-memory) and all of the data features (fast, medium and slowly changing). We then quantified the "timescale-selectivity" — e.g. whether the slow-changing feature was more correlated with long-memory node than other nodes (no-memory and short-memory) (See Figure 3.E).

**Learning Efficiency Measure.** Learning speed was measured using the reconstruction error of the test data, computed as the MSE across all 3 subcomponents of each data sample.

### 5.2.2 RESULTS

We first confirmed that all of the autoencoder (AE) models could learn to reconstruct the input (Figure 3.D). The most efficient architectures were the [ leaky + resetting] AE, [multiscale leaky + resetting] AE, and the memoryless AE.

Both networks with memory and resetting could successfully un-mix fast and slow data sources. The individual hidden state units in these AE models were selectively more correlated with their

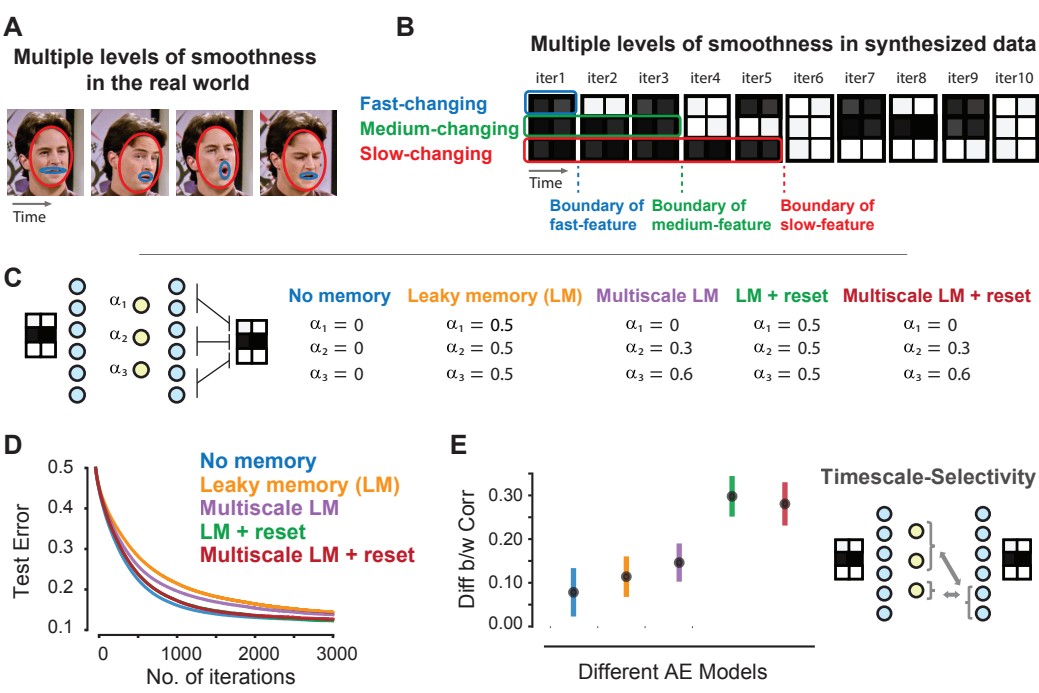

Figure 3: Unsupervised learning from data with multiple levels of smoothness. *A) Example of multiple levels of smoothness in samples from the real world: mouth changing fast, while face-shape changes slowly. B) Multiple levels of smoothness in synthesized data: top row changes every item; middle row changes every 3 items; bottom row changes every 5 items. X-axis shows time, each 3-by-2 item is one sample. C) 5 Different AE models. $\alpha_1$, $\alpha_2$, and $\alpha_3$ show the memory coefficient in the hidden representations. D) Reconstruction test error (MSE loss) during training for individual items across 5 different AE models. All the curves in this plot have been averaged over 50 runs with different random initialization. E) Comparing the "timescale- selectivity" of models, by computing the difference between the squared Pearson correlations for time-scale matching units and non-matching units (e.g. correlation of long-memory with slow features minus correlation of long-memory with fast and medium features). In no-memory systems and in leaky-memory systems with uniform memory, we measured these correlations for hidden units in the corresponding position as those in the multi-scale leaky memory. Error bars show the mean and standard deviation across 10,000 bootstraps, with 50 values per bootstrap.*

corresponding data features (i.e. the slow-changing feature was more correlated with the long-memory node than with the other nodes; Figure 3.E)).

These findings generalized across synthesized datasets and across learning rates (See Appendix A.12).

### 5.2.3 DISCUSSION

The two autoencoder models that had both memory and resetting mechanisms were most successful in learning internal representations that tracked distinct timescales of the input. Slowly (or quickly) varying features were extracted by slowly (or quickly) varying subsets of the network, analogous to a matched filter (see also Mozer (1992)). Features that change on different timescales may correspond to different levels of structure in the world (Wiskott & Sejnowski, 2002). Thus, by adding leaky memory and memory-gating to a simple feedforward AE model, we equipped it with an ability to separate different levels of structure in the environment. Moreover, because intrinsic dynamics vary on multiple scales in the human brain (Stephens et al., 2013; Murray et al., 2014; Honey et al., 2012; Raut et al., 2020), this implies that slowly-varying brain circuits may be biased to extract slowly-varying structure from the world (Honey et al., 2017).

Why did the no-memory (feedforward) model produce slightly lower reconstruction error than models with memory? In models with memory, there is a (small) cost in the overall test error, because slowly-changing internal states are ineffective for reconstructing quickly-changing features. However, the error introduced by nodes reconstructing input from a mismatched timescale is small, and it is accompanied by a significant benefit: learning more meaningful, un-mixed representations of a multi-scale data stream. Indeed, if a model's "slow" hidden units (i.e. medium and long memory units) were correlated with the state fast-changing features in the data, the model's per-feature error was worse (Appendix A.14, Figure A.14).

## 6 CONCLUSION

Inspired by temporal properties of the training signal and the learning architectures in primate brains, we investigated how the smoothness of training data affects incremental learning in neural networks.

First, we examined the speed of learning. We found that data smoothness slowed learning in memoryless learners (feedforward neural nets), but sped learning in systems with leaky memory (Figure 2). Moreover, adding a simple gating mechanism to leaky-memory networks enabled them to flexibly adapt to the smoothness in the data, so that they could benefit from repeating structure while not the interference of unrelated prior information.

Second, we examined the representations learned when unlabeled data contained temporal structure on multiple smoothness levels. Neural networks with memory and feature-sensitive gating learned representations that un-mixed features varying on different timescales. If distinct timescales in data reflect distinct data generators, these "un-mixed" representations may provide a more "meaningful" description of the input data (Mitchell, 2020; Mahto et al., 2020).

Leaky memory networks exhibited more efficient learning and more interpretable representations, even though they were trained with a learning rule that did not employ any temporal information. In particular, all networks were trained incrementally using backpropagation and a loss function that only depended on the immediate state of the network. Architectures with leaky memory and gating can thus exploit temporal structure in a way that is computationally simpler and more biologically plausible than backpropagation through time (Sutskever, 2013; Lillicrap & Santoro, 2019). With respect to biological plausibility, we note that the leaky-memory-plus-gating system works well even for autoencoders (Figure 3), for which there are simple activation-based learning rules that do not require the propagation of partial derivatives (Lee et al., 2015). On the computational side, we highlight that the gradients computed for the leaky memory networks were, in a sense "inaccurate", because the update rule was unaware of the recurrent leak connections, and yet learning in the leaky nets was still faster than in feedforward nets, for which the gradients should be more accurate.

Future work should test whether these results generalize to larger architectures and more realistic datasets, and should include a broader search of the hyperparameter space. The results may have some generality , because we used simple architectures and made few domain-specific assumptions, but at present the results serve as demonstrations of the basic phenomena. We expect the method to work best for datasets in which important or diagnostic data features persist over time. It will also be interesting to investigate the broader consequences of learning with leaky memory: for example, human internal representations of natural sensory input sequences appear to be smooth in time, in contrast to the representations of most feedforward nets (Hénaff et al., 2019)), and training with smooth data and leaky memory could potentially reduce this difference.

In sum, we identified simple mechanisms which enabled neural networks (i) to learn quickly from temporally smooth data and (ii) to generate internal representations that separated distinct timescales of the data, without propagating gradients in time.

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

## A APPENDIX

### A.1 EFFECTS OF SMOOTHNESS ON CATEGORIZATION VERSUS RECONSTRUCTION TASKS

Classification networks (performing categorization) and autoencoder networks (performing reconstruction) were similarly affected by temporal smoothness of training data (Figure A.1). Increased smoothness decreased learning efficiency. Also, "minimum smoothness" sampling exhibited the best performance across both types of networks. We used 3-layer fully connected networks for both classification and reconstruction. The network dimension for classification was (16, 8, 4) and for reconstruction was (16, 4, 16). Learning rate was 0.01 for all conditions in classification, and 0.005 for all conditions in reconstruction.

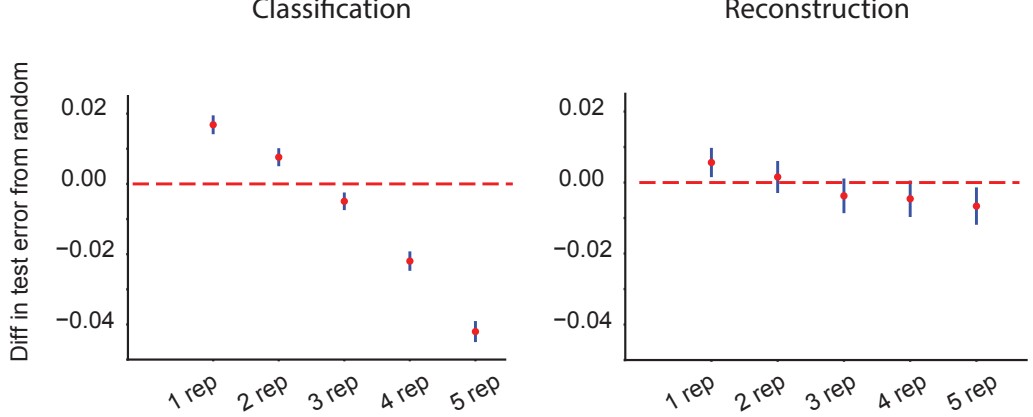

Figure A.1: Comparing the effects of smoothness on classification and reconstruction for a synthetic dataset (Figure A.2). *Left: Difference between the test error with random sampling and the test error in other sampling conditions (e.g. Error(random) - Error(1 repetition)) in the classification task. The dashed line shows the baseline for random sampling. Test errors were computed at the end of the first training epoch. We ran 100 runs with different weight initializations. Error bars show the mean and standard deviation of bootstrapping 10,000 times on 100 values from 100 runs. Right: As for the left panel, but for the reconstruction task.*

### A.2 SYNTHETIC DATASET

We synthesized a dataset with low between-category overlap. The dataset consisted of 4 categories, each with 300 training items. Each item was a 1-by-16 vector. Different exemplars of a category were created by adding uniform noise to the template of the category.

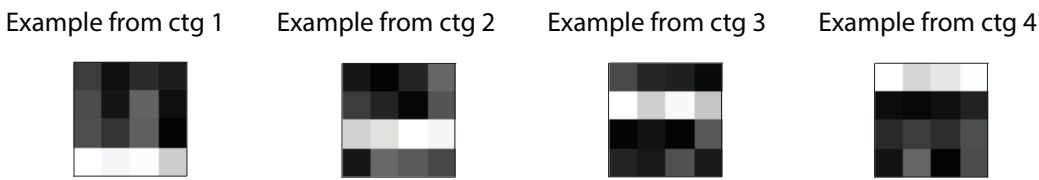

Figure A.2: Example items from each of the 4 categories in the synthetic dataset.

### A.3 SMOOTHNESS EFFECTS FOR CLASSIFICATION USING CROSS-ENTROPY LOSS

Similar to the results obtained with mean-square error (MSE) loss, we found that temporally smooth data slowed category learning with training with cross-entropy (CE) loss. Figure A.3 shows this

effect for the MNIST dataset. We used the same neural architecture and hyperparameters as those in MSE, explained in section 4.1.2.

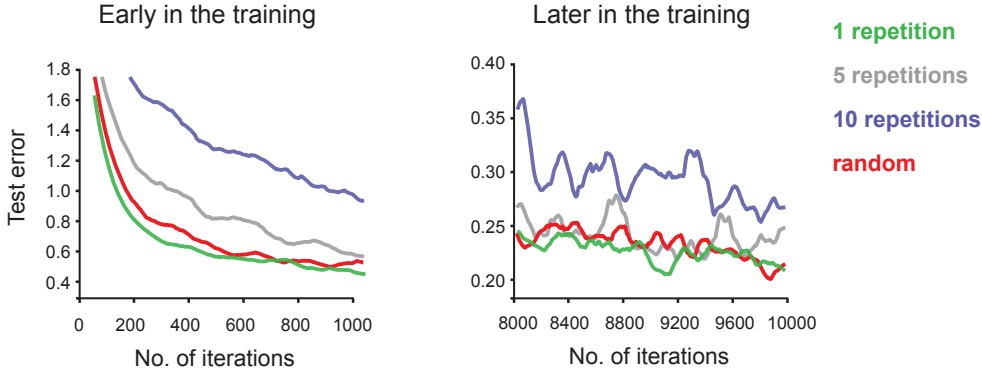

Figure A.3: Test error (CE loss) for SGD training of feedforward neural network on MNIST dataset, early vs later in the training process. *Each color shows a different smoothness condition in the training data.*

## A.4 CATEGORIZATION OF SYNTHETIC DATA BY LEAKY MEMORY NETWORKS WITH GATING

Figure A.4 shows the effects of temporal smoothness in training data for neural network models equipped with leaky memory and gating for the synthetic dataset. Similar to the pattern observed in Figure2.B, we can see that, in the network with leaky memory, higher levels of smoothness generate better performance. Moreover, adding a gating mechanism enhanced learning, such that all levels of smoothness surpassed the "minimum smoothness" (1 repetition) condition, as was observed in Figure2.C.

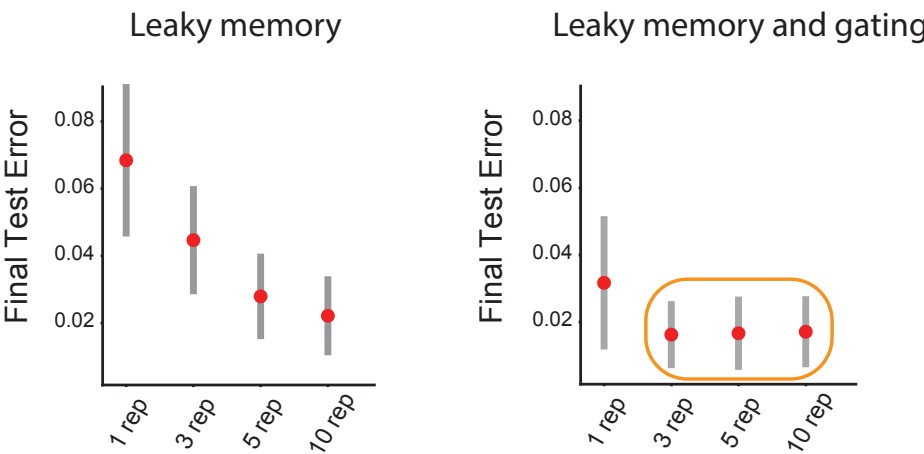

Figure A.4: Effects of temporal smoothness on categorization of synthetic data in networks with leaky memory. *Left: Test error (MSE loss) at the end of first training epoch with SGD on synthetic dataset, for network with leaky memory in internal representation. Right: As for the left panel, but for networks with leaky memory in internal representations and gating. Error bars show the mean and standard deviation of bootstrapping 10,000 times on 100 values from 100 runs with different weight initialization.*

A.5    COMPARING THE LEAKY MEMORY APPROACH AGAINST MINI-BATCH TRAINING

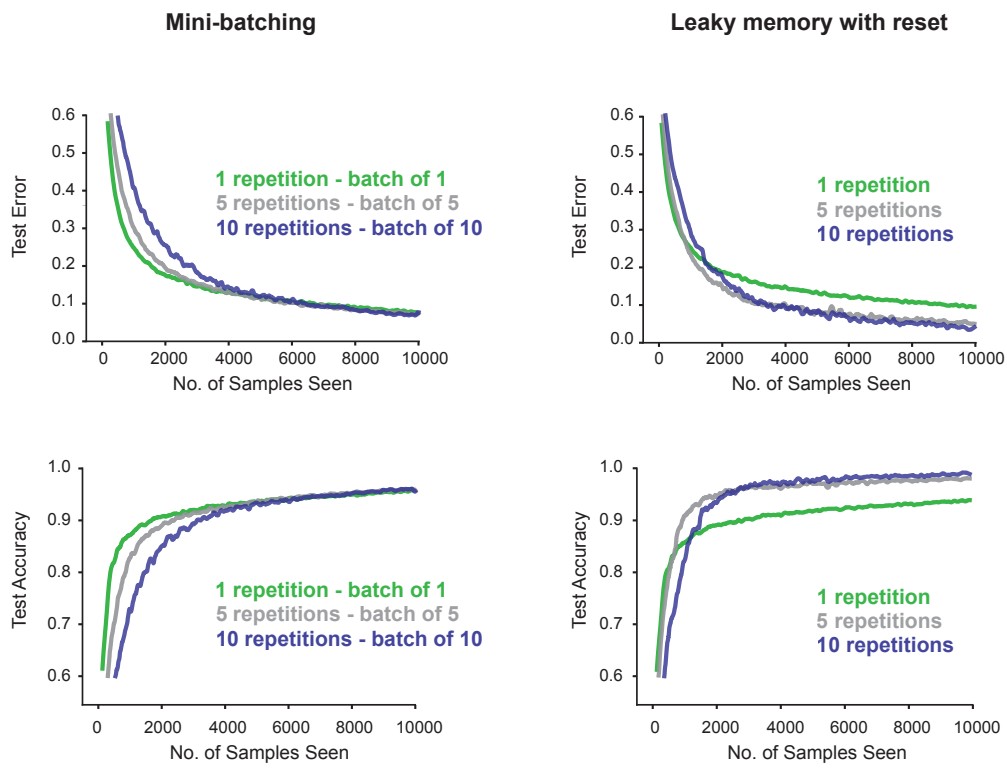

Figure A.5: Mini-batching and [Leaky memory + reset] models are affected in a qualitatively different manner by increasing the number of category repetitions. Left) Test error and test accuracy of mini-batch training on MNIST data. Right) Test error and test accuracy of [leaky memory + reset] model on MNIST data. Both models had the dimension of (784, 392, 10), learning rate of 0.01, and the optimization method of SGD.

A.6    EFFECTS OF SMOOTHNESS IN DATA ON MINI-NATCH TRAINING

We explored how smooth data affects learning when weights were updated using mini-batch training. We used the MNIST dataset and trained each network with batches of size 16. Network dimension and other hyperparameters were identical to those used in incremental SGD. We found that the level of smoothness in the training did not influence mini-batch training similarly to SGD training. Early in the training, minimum smoothness showed the fastest learning and higher levels of smoothness showed slower learning. However, later in the training, another pattern was observed: the condition with the smoothness level equal to the batch size (e.g. 16 repetitions for batch of 16) showed the greatest learning efficiency compared to both lower levels of smoothness (e.g. 10 repetitions) and higher levels of smoothness (e.g. 24 repetitions).

In connecting the mini-batch data to the results reported in Figure 2, consider that "smoothness" can happen at 2 levels: samples can be similar to one another within a batch ("smooth within a batch") and the composition of samples can be similar across consecutive mini-batches ("smooth across batches"). It seems that early in the training, the conditions with minimum "within-batch smoothness" have the highest learning speed; this makes sense as the composition of each mini-batch is most reflective of the overall composition of the test data. However, later in the training, the condition with minimum "across-batch smoothness" has the best learning speed. Minimum

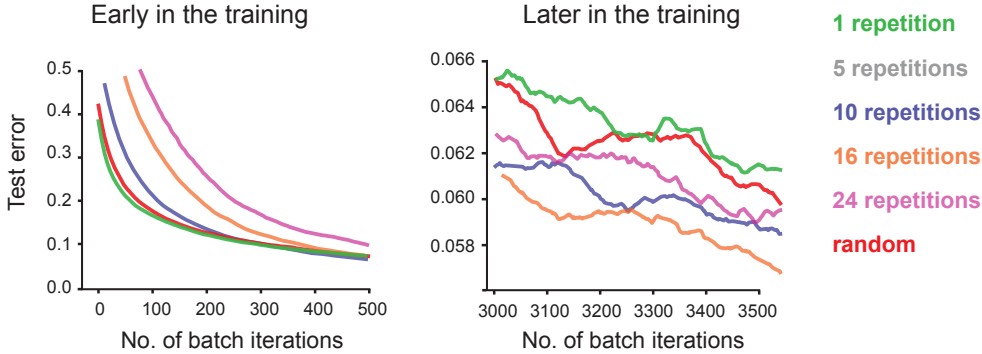

Figure A.6: Left) Test error (MSE loss) in different levels of smoothness in data, early in mini-batch training of MNIST dataset for classification. Right) The same as left, for later in the training, toward the end of first epoch.

across-batch smoothness refers to the condition where each batch consists of items from only one category (e.g. 16 repetitions for batch of 16). Note that when each individual batch contains items all from one category, this also implies that consecutive batches will not contain any items from the same categories, leading to a "minimum smoothness at the batch level". Thus, the advantage for minimum across-batch smoothness may be analogous to what is observed in Figure 2 at the single item level, but occurring at the batch level.

These results are of course, only preliminary, and future work should elaborate how the smoothness of the training data interacts with mini-batch training.

## A.7 Synthetic data streams in which leaky memory is disadvantageous

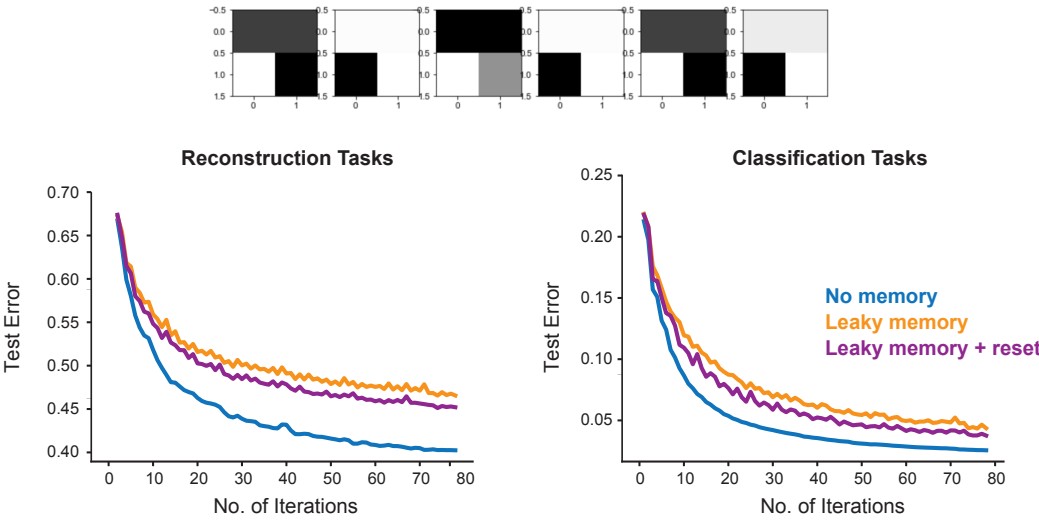

Figure A.7: Synthetic data streams for which leaky memory is disadvantageous. Left) Test error for reconstruction tasks in an autoencoder model. Right) Test error for classification tasks.

We hypothesized that the averaging mechanism in leaky memory models increases the proportional signal variance allocated to category diagnostic features, by emphasizing the features that are shared across multiple members of a category. In order to demonstrate this, we trained our model on a data structure in which the consecutive items did not share any local features.

We used a synthesized dataset and organized the data so that consecutive items did not have shared features. For resetting mechanism, we tried a range of resetting (e.g. every 2 items, every 3 items, etc). In this setting, the leaky memory advantage was eliminated, and leaky memory, with or without reset, was always less effective than learning without memory.

## A.8 Effects of temporal smoothness on category-learning in an LSTM trained with BPTT

We explored how training with smooth data affects category-learning in an LSTM trained with backpropagation through time. The dataset, network dimension, learning rate, and optimization method were identical to ones used in section 5.1.1.

We found that higher smoothness in training data resulted in better classification performance in LSTM.

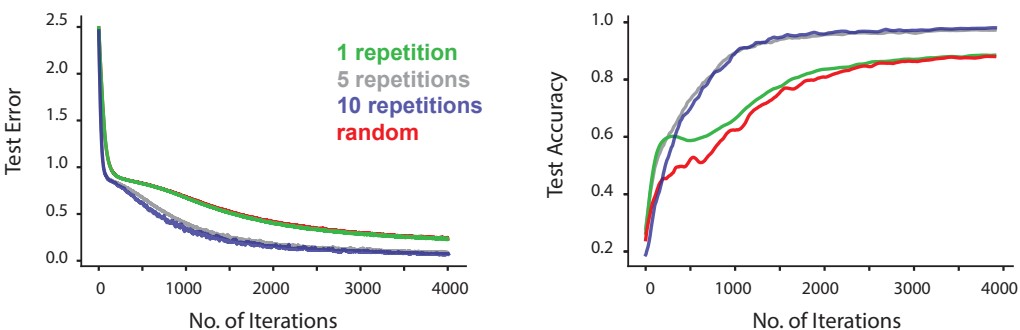

Figure A.8: Effects of smoothness in data on learning LSTM. Left) LSTM test error for different amounts of smoothness in training data. Right) LSTM test error for different amounts of smoothness in training data.

## A.9 Comparing LSTM and [leaky-memory with reset] models

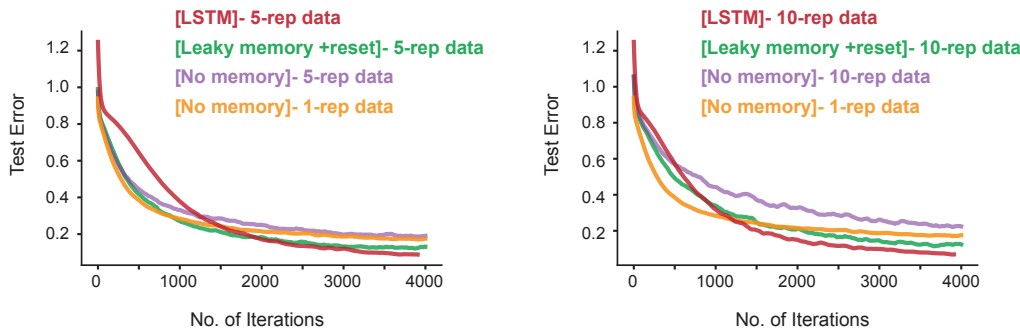

Figure A.9: Comparing LSTM, the [leaky memory + reset] model, and the no-memory model. Left) Test error for classification with smoothness level equal to 5-repetitions of each category. Right) Test error for classification with smoothness level equal to 10-repetitions of each category. In both plots, we also show 1-repetition to be used as a reference.

We compared LSTM, the [leaky memory + reset] model, and the no-memory model. We first tested both the LSTM and the [leaky memory + reset] model on the same data structure that they were

trained on (e.g. training with 5-repetitions and testing on 5-repetitions). This means that we included memory in the testing process and tested the models on the same order that they were trained on. In this setting, LSTM was the best model, and the leaky-memory model was second-best, better than no-memory model.

### A.10 GENERALIZATION OF LSTM AND [LEAKY MEMORY + RESET] MODELS TO DATASETS WITH DIFFERENT TEMPORAL STRUCTURE

We compared LSTM and the [leaky memory + reset] model on their generalizability. To do so, we first trained and tested both models on the same sequence of samples (e.g. trained on 5-repetitions and tested on 5-repetitions). Then we tested them on a different sequence of samples from the one they were trained on (e.g. trained on 5-repetitions and tested on 1-repetition). We found that LSTM outperformed the [leaky memory + reset] model when tested on the same sequence used for training. However, [leaky memory + reset] model was superior to LSTM when tested on a data stream with different temporal structure from training. Differences between LSTM and the [leaky memory + reset] model suggest that the two do not exploit the temporal structure of the data stream in the same way. The LSTM makes use of information about the specific task structure (e.g. there are precisely 5 repetitions in a block) and its performance is reduced when this assumption is violated in generalization data. Conversely, the [leaky memory + reset] model simply uses the temporal smoothness in the training data to learn more useful internal representations.

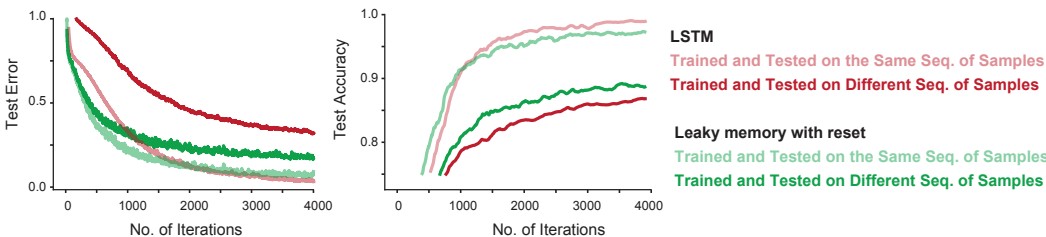

Figure A.10: Generalization of LSTM and leaky-memory models to data streams with different temporal structure. Left) Test error for LSTM and [leaky memory + reset] model, trained and tested on the same sequence of samples or on a different sequence of samples. Right) Test accuracy for LSTM and [leaky memory + reset] model, trained and tested on the same sequence of samples or on a different sequence of samples.

### A.11 UNSUPERVISED LOCAL RESETTING MECHANISM

For learning multiscale data we have implemented a "resetting" mechanism in a straightforward and unsupervised way using only local computations, while preserving the same gains in learning efficiency. To do so, we have used the comparison between the difference and the average of the following input items as the resetting criterion, but other sorts of computations are also possible. Our implemented method is consistent with neurophysiological studies that demonstrate a sudden shift in memory representations in the face of a surprise in the input stimuli (DuBrow et al., 2017; Chien & Honey, 2020). The bioplasible event-related resetting: Reset the memory when the difference between the consecutive inputs is larger than their average. For instance, the memory of the hidden node with long memory will be reset based on the amount of change in the slow-changing feature of the input. [ t represents the iteration number during training, $I_t$ is the current state, $I_{t-1}$ previous state ] $|I_t - I_{t-1}| > |(I_t + I_{t-1})/2|$

### A.12 GENERALIZABILITY OF FINDING IN LEARNING FROM MULTISCALE DATA FOR A DIFFERENT LEARNING RATE AND A DIFFERENT DATASET

To investigate the generalizability of the findings from section 5.2, we examined the performance of the model for a range of learning rates. Part A in figure A.12 shows the results for learning rate of 0.003, in the same dataset reported in section 5.2.2. To investigate the generalizability of the findings

from section 5.2, we examined the performance of the model for different synthesized datasets. Part B in Figure A.12 shows the results for a different synthesized dataset (learning rate = 0.005).

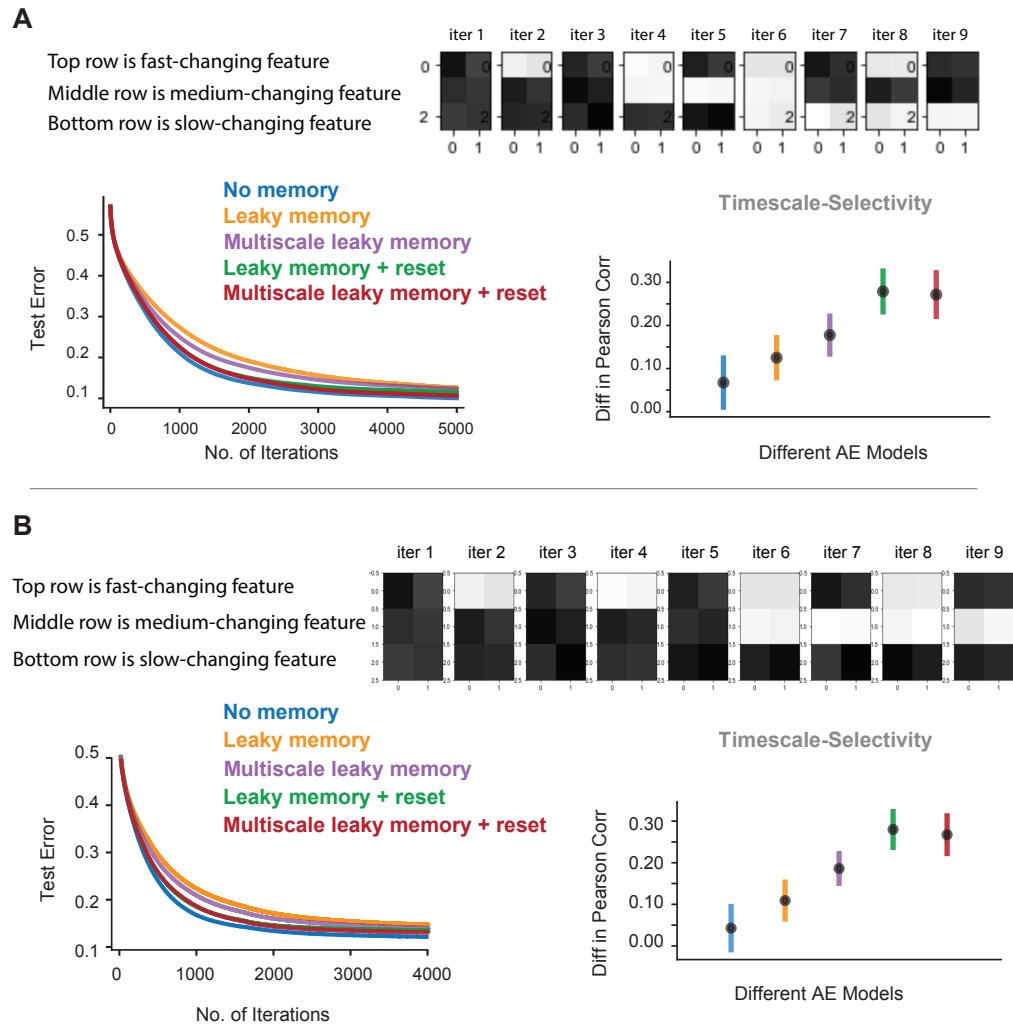

Figure A.12: A) Learning performance and internal representations of autoencoder models with and without memory for learning rate of 0.003 in the same dataset used in Figure 4. (See Figure 4 for more details.) B) Learning performance and internal representations of autoencoder models with and without memory for a different dataset from the one used in Figure 4.

## A.13 DOES FASTER CONVERGENCE IN THE NO-MEMORY MODEL FROM 5.2 CONTRADICT THE BENEFIT OF THE MEMORY-RESET MODEL FROM 5.1?

The findings from section 5.1 and 5.2 are complementary rather than contradictory. Consider that the multiscale data stream in part-2 is composed of three different subcomponents (top, middle and bottom rows of the input). Thus, the multi-scale stream can be understood as a combination of the 1-rep condition from part-1 (feature changes with every sample), the 3-rep condition from part-1( feature changes at a medium speed across samples), and the 5-rep condition from part-1 (the feature changes slowly across samples).In part-1, we showed that the [leaky memory + reset] model performs better when the data has higher smoothness (e.g. in Figure 2.C, category learning is more efficient for 5-repetitions than for 1-repetition). If the same pattern holds in the context of multi-scale autoencoders, for models with memory and reset, we should see that the slow-changing

feature (5-repetitions) is more quickly learned than the fast-changing feature (1-repetition). To test this, we measured the per feature error (e.g. test error for reconstructing a specific feature) and we compared the fast and slow-changing features. Consistent with the pattern observed in part-1 of the paper, we saw that both the [leaky memory + reset] model and [multiscale leaky memory + reset] model exhibited the predicted pattern ( Figure A.13): the reconstruction error for the slow changing feature was lower than the reconstruction error for the fast-changing feature.

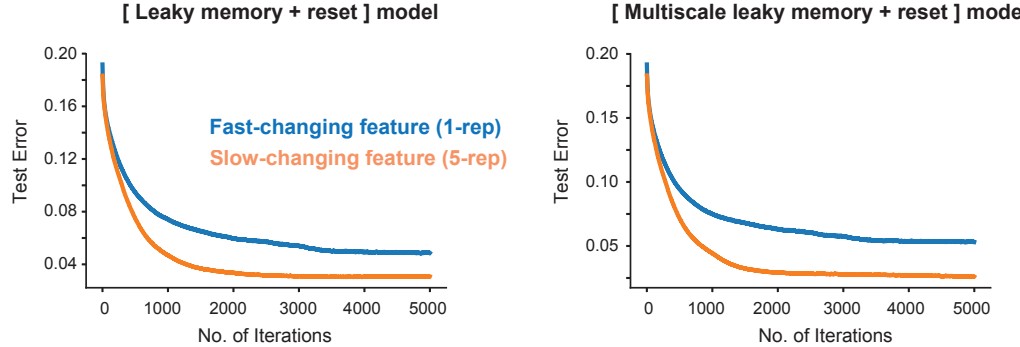

Figure A.13: Per feature error for fast-changing and slow-changing features. Left) Feature-wise test error for [leaky memory + rest] model. Right) Feature-wise test error for [multiscale leaky memory + reset] model.

## A.14 WHY DOES THE [NO-MEMORY] MODEL OUTPERFORM THE [MULTISCALE MEMORY + RESET] MODEL IN TEST-ERROR (IN SECTION 5.2)

The multi-scale model faces a challenge that was not faced by the single-scale models that were tested in the category learning components of this study. Because all nodes in the hidden layer of the multi-scale model project to all nodes in the reconstruction later, the slowly changing hidden states of the model (i.e. the nodes with longer memory) are contributing to the reconstruction of quickly-changing features in the data stream. There is a (small) cost in the overall test error, because slowly-changing internal states are ineffective for reconstructing quickly-changing features. We emphasize that the quantity of noise introduced is small, and that it is accompanied by a significant benefit in learning more interpretable, un-mixed representations of a multi-scale datastream. To demonstrate that these slow units are indeed the source of poorer learning, we tested the hypothesis that (i) a higher correlation between hidden units with memory (units with short or long memory) and fast-changing part of the output would result in a worse performance in reconstructing the fast-changing feature; whereas (ii) a higher correlation between hidden unit with no-memory and fast-changing part of the output would not result in worse performance in reconstructing the fast-changing feature. These hypotheses were confirmed in our analyses [Figure A.14].

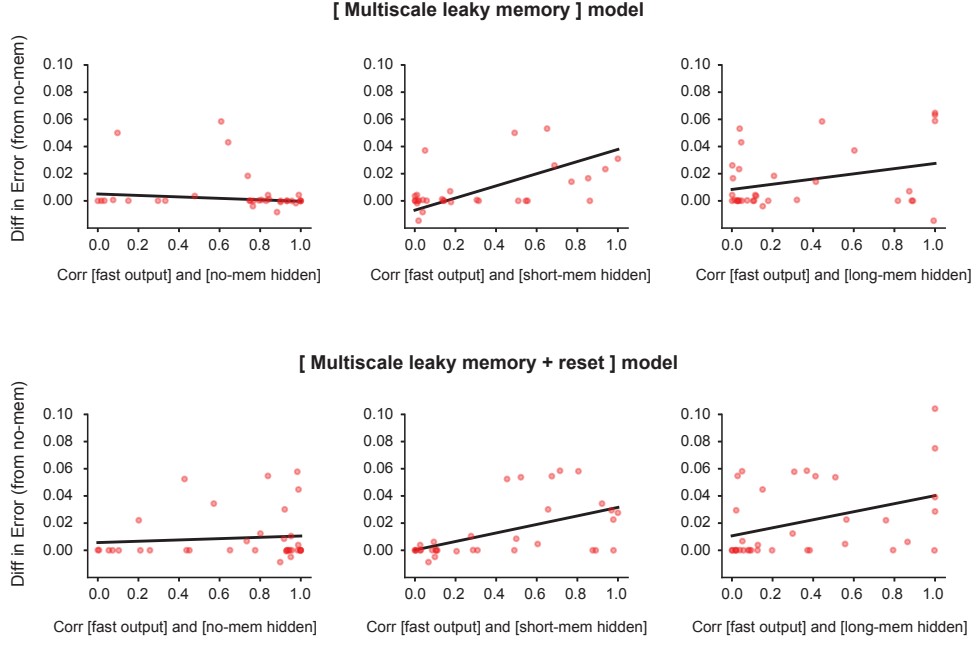

Figure A.14: Higher correlation between slow hidden nodes and fast-changing feature of the output resulted in worse performance in reconstructing fast-feature.

A.15    DYNAMICS OF HIDDEN UNITS IN DIFFERENT AUTOENCODER MODELS THAT ARE
         LEARNING TO RECONSTRUCT MULTI-TIMESCALE INPUTS

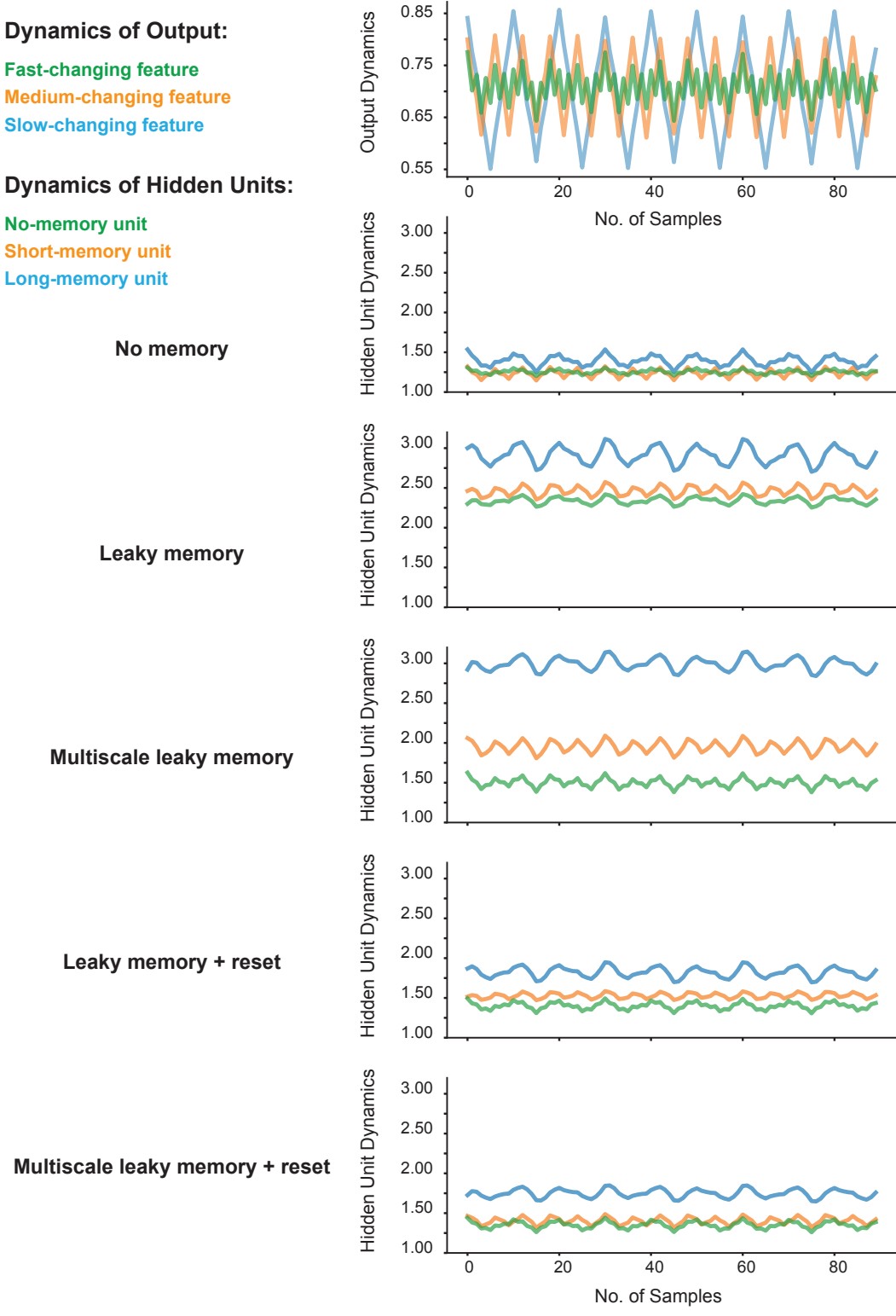

Figure A.15:   Visualization of dynamics of hidden units from different AE models on the held out data.

### A.16 OPEN-SOURCE

The source code will be shared on authors' github after the reviewing process.

