# OpenReview forum: "Learning representations from temporally smooth data"
_ICLR.cc/2021/Conference — Reject_

### Official Review · AnonReviewer3 · 2020-10-27
**This is an empirical paper which shows that leaky memory and memory-gating can take advantage of temporal smoothness in data. They run multiple experiments to show the effectiveness. However, one of my main concern is that such a mechanism may be hard to apply in practical scenarios in particular with class imbalance.**

**Rating:** 6
**Confidence:** 3

**Review:**

In regular DNN training mini-batches are selected at random and temporal smoothness of data is not used. In fact it is expected that temporal smoothness can lead to catastrophic forgetting which may lead to poorer performance. The authors of this paper first verify this hypothesis and prove this to be true.

Then they propose that two memory inspired mechanisms can take advantage of the temporal smoothness of data: leaky memory and memory-gating. They show that leaky memory helps when data is presented smoothly and additional gating helps even further (however, the legends in Figure 2 are not clear). They also show that similar results hold for unsupervised learning.

While the results are promising (and I also liked the link the authors draw from human brain to the two proposed mechanisms) there are certain drawbacks as well. One of the main advantages of random training is that generating the order is not costly. However, in the temporarily smooth approach this can become a bottleneck due to the large size of datasets used to train DNNs. Another potential issue pops up due to data imbalance found in practice. While undersampling/oversampling can be used, data imbalance can still be a big problem. In that respect I would have preferred to have seen experiments with more datasets and more complex neural networks.

Comments:
- Show the legends for all curves in Figure 2 (only Minimum smoothness is marked as green)
- Why is the random sampling in 2B different from 2A and 2C? They should all be the same or just different. Any particular reason for treating 2B differently?
- I would suggest showing cross-entropy results in the main paper as that is the popular loss function used.
- I am confused by 4A. It seems that No memory has the lowest test error. Doesn't that mean no memory is the best?
- Typo: Related Work: line 2: speeded -> sped

---

> ### Author Response · Authors · 2020-11-25
> **Our Response to Reviewer 3 (Part 1)**
>
> We thank the reviewer for the positive and supportive feedback.
>
> * Comment: While the results are promising (and I also liked the link the authors draw from human brain to the two proposed mechanisms) there are certain drawbacks as well. One of the main advantages of random training is that generating the order is not costly. However, in the temporarily smooth approach this can become a bottleneck due to the large size of datasets used to train DNNs. Another potential issue pops up due to data imbalance found in practice. While undersampling/oversampling can be used, data imbalance can still be a big problem. In that respect I would have preferred to have seen experiments with more datasets and more complex neural networks.
>
> Thanks for pointing this out.
>
> Regarding data imbalance, we now have added some discussion into section “Related Work” about the iid assumption in machine learning research. Although we agree that random sampling has its own advantageous due to homogeneity, it is also very different from how humans perceive information from the world. As Hadsell et al. (2020) recently argued:
>
> “Modern machine learning excels at training powerful models from fixed datasets and stationary environments, often exceeding human-level ability. Yet, these models fail to emulate the process of human learning, which is efficient, robust, and able to learn incrementally, from sequential experience in a non-stationary world.”
>
> Regarding investigating generalization to more datasets and more complex neural networks, this is an important next step for our work, which we are currently addressing by training our models on the e-vds dataset which shows short video clips of everyday objects, such as shoes, plants, and cups.
>
> Moreover, while we are working to extend to more realistic datasets, the toy datasets provide us with control necessary to determine some of the mechanisms and principles -- without toy data, we would not be able to perform manipulations like the ones shown in Appendix A.7, which helps to reveal the mechanism. Also, we are not only motivated to develop new machine learning approaches, but also seek to understand how learning unfolds in the human brain. From this perspective, we believe that the use of simplified datasets and generic architectures is not a drawback.
>
> * Comment: Show the legends for all curves in Figure 2 (only Minimum smoothness is marked as green)
>
> Thanks for pointing this out and we apologize for the confusion. We have modified Figure 1 in our revised paper, so that all curves are similarly labeled.
>
>
> * Comment: Why is the random sampling in 2B different from 2A and 2C? They should all be the same or just different. Any particular reason for treating 2B differently?
>
> This is an interesting point, thanks for mentioning it. In the original paper, random-sampling in Figure 2B has memory, which made it different from random sampling in 2A and 2C. In the revised paper, the random sampling curve in all 3 plots is identical and can be used as a common reference.
>
> * Comment: I would suggest showing cross-entropy results in the main paper as that is the popular loss function used.
>
> Thanks for pointing this out. As we reported in the paper, we used MSE, primarily for the ease of comparison with later reconstruction error measures in this manuscript. However, the same pattern was observed using CE loss, as shown in Appendix A.3. Also, it has been shown MSE loss provides comparable performance to commonly utilized classification models with CE loss function (Illing et al., 2019).

---

> > ### Author Response · Authors · 2020-11-25
> > **Our Response to Reviewer 3 (Part 2)**
> >
> > * Comment: I am confused by 4A. It seems that No memory has the lowest test error. Doesn't that mean no memory is the best?
> >
> > Thanks for pointing this out. We have looked into this more closely, and we believe that this was helpful for better understanding how these multi-scale systems operate. Overall, our analyses indicate that the findings from Section 5.1 and Section 5.2 are complementary rather than contradictory.
> >
> > Does faster convergence in the no-memory model from Section 5.2 contradict the benefit of the memory-reset model from Section 5.1?
> >
> > We do not believe that the results are contradictory. Consider that the multiscale data stream in Section 5.2 is composed of three different subcomponents (top, middle and bottom rows of the input). Thus, the multi-scale stream can be understood  as a combination of the 1-rep condition from Section 5.1 (where the feature changes with every sample), the 3-rep condition from Section 5.1(where the feature changes at a medium speed across samples), and the 5-rep condition from Section 5.1 (where the the feature changes slowly across samples).
> >
> > In Section 5.1, we showed that the [leaky memory + reset] model performs better when the data has higher smoothness (e.g. in Figure 2.C, category learning is more efficient for 5-repetitions than for 1-repetition). If the same pattern holds in the context of multi-scale autoencoders, for models with memory and reset, we should see that the slow-changing feature (5-repetitions) is more quickly learned than the fast-changing feature (1-repetition). To test this, we measured the per feature error (i.e. test error for reconstructing a specific feature) and we compared the fast and slow-changing features. Consistent with the pattern observed in section 5.1 of the paper, we saw that both the [leaky memory + reset] model and [multiscale leaky memory + reset] model exhibited the predicted pattern ( See Figure A.13): the reconstruction error for the slow changing feature was lower than the reconstruction error for the fast-changing feature.
> >
> > Why does the [no-memory] model outperforms the [multiscale leaky memory + reset] model in test-error, in Section 5.2?
> >
> > The multi-scale model faces a challenge that was not faced by the single-scale models in Section 5.1: because all nodes in the hidden layer project to all nodes in the reconstruction later, the slowly changing hidden states of the model (the ones with longer memory) are contributing to the reconstruction of quickly-changing features in the data stream. There is a (small) cost in the overall test error, because slowly-changing internal states are ineffective for reconstructing quickly-changing features. We emphasize that the quantity of noise introduced is small, and that it is accompanied by a significant benefit in learning more interpretable, un-mixed representations of a multi-scale datastream. Nonetheless, this is an important trade-off, which we now demonstrate in the Appendix and in Figure A.14.
> >
> > To demonstrate  that “interference” from  slow units is indeed the source of poorer learning, we tested the hypothesis that (i) a higher correlation between hidden units with memory (units with short or long memory) and fast-changing part of the output would result in a worse performance in reconstructing the fast-changing feature; whereas (ii) a higher correlation between hidden unit with no-memory and fast-changing part of the output would not result in worse performance in reconstructing the fast-changing feature. These hypotheses were confirmed in our analyses shown in Figure A.14.
> >
> >
> > * Comment: Typo: Related Work: line 2: speeded -> sped
> >
> > Thanks for kindly pointing this out. We modified it in the revised paper.

---

### Official Review · AnonReviewer2 · 2020-10-27

**Rating:** 4
**Confidence:** 4

**Review:**

**Update after rebuttal:** I appreciate the detailed responses by the authors. I'm willing to increase my score based on the responses, but unfortunately I'm still not ready to recommend acceptance. In my opinion, the paper is simply not mature enough yet for publication (the significant amount of revisions required during the rebuttal period attests to this, I think; a mature conference paper should not have to require this much revision during review). In particular, the following fundamental issues still remain for me even after the revisions:

1. The misleading language about "temporal smoothness" in real-word data remains throughout the paper despite the fact that the paper doesn't address temporal smoothness as it exists in real-world data.

2. The authors promise some new experiments on more realistic stimuli, but as it stands the paper still only includes experiments on static images with mostly toy data and I have no way of knowing whether any of their results would generalize to more realistic data. The experiments with multi-scale stimuli suggest that that generalization may be non-trivial (e.g. in that experiment, the baseline model with no memory or gating mechanisms actually performs the best).

3. Which brings me to my final point: I still don't think the authors have adequately explained why and how the proposed mechanisms work. For example, the authors say: *"Our working hypothesis is that averaging across multiple members of the same category increases (in some datasets) the proportion of variance in the hidden units that is associated with category-diagnostic features."* Why the hedging *in some datasets*? The experiments with multi-scale stimuli clearly demonstrate that the proposed scheme doesn't work in all cases, but what exactly are the conditions under which it would work better than the baseline model? The authors need to make these a lot clearer.

------------------------------------------
This paper mainly investigates the effect of iteration-to-iteration correlations in online learning. It recapitulates a fairly obvious and pretty well-known result that such iteration-to-iteration correlations will slow learning. I find the motivating question (the effect of temporal correlations on learning) somewhat interesting, but unfortunately, I think the research reported in this paper is really not very well-executed:

(1) Unlike what the title and sections 1 and 2 claim, the experiments in this paper do not test the effect of temporal smoothness. There is no actual time dimension in the data used in this paper. It rather tests something else: more accurately described as “iteration-to-iteration correlations in online learning”. This makes the set-up considerably less general and less interesting in my mind compared to the actual temporal ordering question, which has some practical relevance. Relatedly, the illustration in Figure 1A is misleading. This is not the setup tested in the experiments.

(2) The models and datasets used in this paper are extremely toy, there is no reason why more realistic datasets with actual temporal structure could not be used for this research.

(3) The paper only studies the online learning scenario (Appendix A5 reports the results of an experiment with minibatch training, but this is very limited, and not nearly rigorous enough). This limits the relevance of this work both for machine learning and for neuroscience/psychology. Most machine learning research does not do online learning. Even animals do not have to do purely online learning, because they have offline replay mechanisms that don’t have to respect temporal order strictly.

(4) The authors propose two mechanisms to alleviate the learning slow-down caused by iteration-to-iteration correlations in online learning. However, it isn’t at all clear why the proposed mechanisms help with correlated data. No explanation is given for how these mechanisms are supposed to help with learning from correlated data in the online setting. Please note claiming that these mechanisms are brain-inspired is not an explanation. Moreover, the set-up in these experiments is also not described clearly. Section 5.1.1 says “The learning algorithm, optimization and initialization methods, and the hyperparameters were identical to those used in training and testing feedforward neural networks”, but you can’t do online learning with leaky neurons anymore. Later on (right at the very end of the paper in the Conclusion section!), we learn that the learning setup is actually not identical: backprop is truncated in these models to prevent gradients from flowing into previous time steps. This important detail is somehow never mentioned in section 5.

(5) Is it possible that the effect of leaky memory is just due to reduced gradient variance via some sort of mini-batching mechanism? (note that Appendix A5 doesn’t address this question). Since the hidden state contains information about previous examples in this model, the memory may be acting as some sort of implicit mini-batching mechanism that reduces the gradient variance.

(6) The results in Figure 4A: the baseline no-memory model is outperforming the other models. This seems to contradict the results earlier in the paper (e.g. Figure 2) showing the benefits of memory+gating. What is the explanation for this discrepancy?

(7) The experiments are also in general not done very rigorously. For example, no hyperparameter tuning was done for the “smooth” case, but maybe the problem is just that the learning rate in this case should be slightly different (i.e. no need for special mechanisms like memory or gating). We can never know this unless the experiments are done more rigorously.

---

> ### Author Response · Authors · 2020-11-25
> **Our Response to Reviewer 2 (Part 1)**
>
> We thank the reviewer for their detailed feedback. We have revised the manuscript and look forward to their thoughts.
>
> * Comment: Unlike what the title and sections 1 and 2 claim, the experiments in this paper do not test the effect of temporal smoothness. There is no actual time dimension in the data used in this paper. It rather tests something else: more accurately described as “iteration-to-iteration correlations in online learning”. This makes the set-up considerably less general and less interesting in my mind compared to the actual temporal ordering question, which has some practical relevance. Relatedly, the illustration in Figure 1A is misleading. This is not the setup tested in the experiments.
>
> We agree with the reviewer that the autocorrelation structure we are testing here does not match the autocorrelation structure of real-world visual information. This is an important next step for our work, which we are currently addressing by training our models on the “e-vds” dataset which shows short video clips of everyday objects, such as shoes, plants, and cups.
>
> That said, we do not agree that the setting in which we have addressed the problem is less interesting or less general than the vision case. In fact, the problems of temporal autocorrelation in the setting of spatiotemporal-vision are quite task-specific: for example, one needs to employ a convolutional architecture and the temporal autocorrelation structure will reflect object motion as well as the relative motion of foreground and background features. It is unlikely that these forms of autocorrelation structure will generalize to other modalities (e.g. odor, audition), and certainly not to other levels of abstraction (e.g. learning of abstract semantic categories or situational schemas). Therefore, in our initial investigations, we have focused on a generic architecture and simplified stimulus sets in order to determine the basic principles, before applying these approaches to more particular cases with more immediate real-world applicability.
>
> We are not only motivated to develop new machine learning approaches, but also seek to understand how learning unfolds in the human brain. From this perspective, the use of simplified datasets and generic architectures is not a drawback. The fact that learning can be accelerated using a simple leak mechanism, without any recourse to backpropagation through time (BPTT), is significant, because BPTT is a biologically implausible approach. Also, in our revisions to the paper, we have now shown that the “resetting” mechanism can be implemented in a biologically-plausible  unsupervised way using only local computations, while preserving the same gains in learning efficiency [see Appendix A.11].
>
> Regarding the illustration in Figure 1.A, we apologize for the confusion. We meant to imply that data in the real world are correlated across nearby points in time, whereas data in training neural networks for categorization or reconstruction tasks are commonly randomized. In “smooth information in the real world” in Figure 1.A, different angles of the same face were supposed to represent different samples of the same category (i.e. each person being a different category). In “random sampling in training neural networks” in Figure 1.A, the different faces were meant to represent shuffled samples from all categories. In the revised version of the paper, we modified Figure 1.A to make our point more clear and avoid confusion.
>
>
> * Comment: The models and datasets used in this paper are extremely toy, there is no reason why more realistic datasets with actual temporal structure could not be used for this research.
>
> We agree with the reviewer that real-world test cases are desirable. We are currently working on extending these results to the case of categorizing everyday objects from video data. As we noted in our response to Point (1) above, we think that the present results are interesting in their own right, both because they hold for a generic architecture (not optimized for a particular modality or task-set) and because their simplicity renders them more amenable to implementation in biological hardware.
>
> Moreover, while we are working to extend to more realistic datasets, the toy datasets provide us with control necessary to determine some of the mechanisms and principles. Without toy data, we would not have been able to perform manipulations like the ones shown in Appendix A.7,  which revealed that the leaky-memory method only works when items within the same category contain overlapping features.

---

> > ### Author Response · Authors · 2020-11-25
> > **Our Response to Reviewer 2 (Part 2)**
> >
> > * Comment: The paper only studies the online learning scenario (Appendix A5 reports the results of an experiment with minibatch training, but this is very limited, and not nearly rigorous enough). This limits the relevance of this work both for machine learning and for neuroscience/psychology. Most machine learning research does not do online learning. Even animals do not have to do purely online learning, because they have offline replay mechanisms that don’t have to respect temporal order strictly.
> >
> > We agree that understanding the interaction between leaky memory and batching is an important step. In the revised paper, we have added new analyses to Appendix A.5 to make clearer that the two phenomena (batching and leaky memory) are different [Please also see our response to Point (5)].
> >
> > Regarding the comment that the online learning case is not relevant to the machine learning community: we agree that most machine learning research does not examine online learning, but this does not mean that it is not an important area. As Hadsell et al (2020) recently argued:
> >
> > “Modern machine learning excels at training powerful models from fixed datasets and stationary environments, often exceeding human-level ability. Yet, these models fail to emulate the process of human learning, which is efficient, robust, and able to learn incrementally, from sequential experience in a non-stationary world.”
> >
> > Regarding the comment on the neuroscience side: although offline replay mechanisms mean that learning does not have to respect temporal order strictly, it is important to clarify two critical points:
> >
> >  First, the autocorrelation of brain states is a large and robust empirical phenomenon [Stephens et al., 2013; Murray et al., 2014; Honey et al., 2012; Raut et al., 2020] -- so that even the brain states that are stored temporarily and later analyzed via replay are already affected by this property of the neural dynamics. Thus, understanding how learning systems work in the setting of autocorrelated data and autocorrelated internal states is critical for understanding the neural processes, regardless of whether the learning process must respect the temporal order.
> >
> > Second, while there is a conceptual connection between offline replay processes and mini-batch training, it is a very indirect relationship. This is because, in the brain, items selected for replay are not a representative sample of the training data -- they are biased by task and reward relevance [Wu and Foster, 2014]. Moreover, weight updates (synaptic change) in the brain are not restricted to replay epochs (i.e. weights are not updated in step with batching) [Wu and Foster, 2014].
> >
> > In sum, the intrinsic autocorrelation of neural dynamics (which arises due to intrinsic circuit-level constraints) is a prominent feature of brain dynamics, and the influence of this property on learning in the human brain is an important problem in its own right. In particular, it is important to understand how the temporal autocorrelation of brain dynamics will aid or impede learning from data-streams that are temporally random or temporally correlated. We have attempted to revise the Introduction to more clearly articulate this motivation.

---

> > > ### Author Response · Authors · 2020-11-25
> > > **Our Response to Reviewer 2 (Part 3)**
> > >
> > > * Comment: The authors propose two mechanisms to alleviate the learning slow-down caused by iteration-to-iteration correlations in online learning. However, it isn’t at all clear why the proposed mechanisms help with correlated data. No explanation is given for how these mechanisms are supposed to help with learning from correlated data in the online setting. Please note claiming that these mechanisms are brain-inspired is not an explanation. Moreover, the set-up in these experiments is also not described clearly. Section 5.1.1 says “The learning algorithm, optimization and initialization methods, and the hyperparameters were identical to those used in training and testing feedforward neural networks”, but you can’t do online learning with leaky neurons anymore. Later on (right at the very end of the paper in the Conclusion section!), we learn that the learning setup is actually not identical: backprop is truncated in these models to prevent gradients from flowing into previous time steps. This important detail is somehow never mentioned in section 5.
> > >
> > >
> > > We agree with the reviewer that the paper would be strengthened by providing more insight as to why the mechanisms work, and we have run new analyses and revised the paper to address this (see below).
> > >
> > > We apologize for the confusion regarding BPTT. In the original version of the paper, it was only in the Discussion that we emphasized the importance of training without BPTT. In the revised version of the paper, we now make clear in the Introduction that one of our objectives is to perform learning without requiring gradients to flow into prior timepoints, because we seek simple and biologically plausible approaches. Indeed, one of the primary findings is that learning with leaky memory units can be more efficient than training without memory, even though the backprop gradients are mathematically derived for the no-memory case, not for the leaky-memory case.
> > >
> > > There appears to be some misunderstanding in relation to this claim: “you can’t do online learning with leaky neurons anymore”. One of the primary demonstrations of this project is that one can do online learning with leaky neurons. It works, even when the gradient computation does not account for the fact that the neurons are leaky.
> > >
> > > Why do the proposed mechanisms help?
> > >
> > > First, in the case of the reset mechanism, the answer is as following: in the presence of leaky neurons, the reset mechanism reduces the interference between information from unrelated training data. Training the category “trousers” on an image that is a mixture of trousers and shoes is simply inferior to training on trouser images. The reset mechanism prevents this cross-category interference.
> > >
> > > Second, why does simple weighted averaging of the current and prior states produce more efficient learning from sequentially-correlated data streams? Our current hypothesis is that the averaging increases the proportional signal variance allocated to category-diagnostic features, by emphasizing the features that are shared across multiple members of a category. In order to demonstrate this, we trained our model on a data structure in which the consecutive items  do not share any local features. In this setting, the leaky memory advantage was eliminated, and leaky memory was always less effective than learning without memory (See Appendix A.7). In this sense, we think of the leaky memory as a form of inductive bias, in which the model assumes that some task-relevant features are persistent over consecutive samples; when the data are consistent with this inductive bias, learning is accelerated.

---

> > > > ### Author Response · Authors · 2020-11-25
> > > > **Our Response to Reviewer 2 (Part 4)**
> > > >
> > > >  * Comment: Is it possible that the effect of leaky memory is just due to reduced gradient variance via some sort of mini-batching mechanism? (note that Appendix A5 doesn’t address this question). Since the hidden state contains information about previous examples in this model, the memory may be acting as some sort of implicit mini-batching mechanism that reduces the gradient variance.
> > > >
> > > >
> > > > This is an interesting question, which was the original motivation for our inclusion of the mini-batching effects shown in Appendix A.6 in the revised paper (Appendix A.5 in the original paper).
> > > >
> > > > Certainly, mini-batching is conceptually similar, because in one case the gradients are averaged across training samples, and in the other case the activation patterns themselves are averaged (weighted average).
> > > >
> > > > We have added new analysis to Appendix A.5 to make absolutely clear that the two phenomena (batching and leaky memory) are not equivalent.
> > > >
> > > > First, in Figure A.5, we now show that batching and memory-reset exhibit qualitatively different relationships with the smoothness of the data. Batching of size 10 with 10-rep data performs worse than a batching of size 1 with 1-rep data. In contrast, in the [leaky memory + reset] model, 10-rep performs better than 1-rep. Moreover, in batching, different batch sizes eventually converge to the same level of performance. However, in the [leaky memory + reset] model, higher repetitions (higher smoothness) results in improved performance. We should note that the x-axis in these figures shows the number of samples seen by the models, rather than the number of training iterations. Therefore, for the same number of samples, there are fewer updating iterations in mini-batching, and thus a shorter run time, which is one of the main reasons for the desirability of mini-batch training. However, given the same number of training samples, the weighted-averaging of hidden states in the memory-reset case results in a lower generalization error compared to averaging gradients in the mini-batching. Having said that, the purpose of this comparison is not to demonstrate the advantage of memory-reset over mini-batching, because the two serve different purposes. Instead, we hope this analysis shows that mini-batching and leaky memory  are qualitatively different.
> > > >
> > > > Second, when our model was trained on a data structure in which consecutive samples did not share local features, memory advantage was eliminated (See Appendix A.7). In contrast, mini-batching would work just fine under this circumstance. To see the difference between leaky memory and batching directly, consider a simplified example.   For batching at size 6, the sequence of training samples ABABAB and AAABBB are exactly equivalent, producing the identical gradient. However, in the case of leaky memory, the sample sequence  AAABBB and ABABAB can generate totally different gradient sequences. For example, in the case where A = -B, there will be  dramatic interference between consecutive stimuli in the ABABAB case, but much less interference in the AAABBB case.

---

> > > > > ### Author Response · Authors · 2020-11-25
> > > > > **Our Response to Reviewer 2 (Part 5)**
> > > > >
> > > > > * Comment: The results in Figure 4A: the baseline no-memory model is outperforming the other models. This seems to contradict the results earlier in the paper (e.g. Figure 2) showing the benefits of memory+gating. What is the explanation for this discrepancy?
> > > > >
> > > > > Thanks for pointing this out. We have looked into this more closely, and we believe that this was helpful for better understanding how these multi-scale systems operate. Overall, our analyses indicate that the findings from Section 5.1 and Section 5.2 are complementary rather than contradictory.
> > > > >
> > > > > Does faster convergence in the no-memory model from Section 5.2 contradict the benefit of the memory-reset model from Section 5.1?
> > > > >
> > > > > We do not believe that the results are contradictory. Consider that the multiscale data stream in Section 5.2 is composed of three different subcomponents (top, middle and bottom rows of the input). Thus, the multi-scale stream can be understood  as a combination of the 1-rep condition from Section 5.1 (where the feature changes with every sample), the 3-rep condition from Section 5.1 (where the feature changes at a medium speed across samples), and the 5-rep condition from Section 5.1 (where the the feature changes slowly across samples).
> > > > >
> > > > > In Section 5.1, we showed that the [leaky memory + reset] model performs better when the data has higher smoothness (e.g. in Figure 2.C, category learning is more efficient for 5-repetitions than for 1-repetition). If the same pattern holds in the context of multi-scale autoencoders, for models with memory and reset, we should see that the slow-changing feature (5-repetitions) is more quickly learned than the fast-changing feature (1-repetition). To test this, we measured the per feature error (i.e. test error for reconstructing a specific feature) and we compared the fast and slow-changing features. Consistent with the pattern observed in section 5.1 of the paper, we saw that both the [leaky memory + reset] model and [multiscale leaky memory + reset] model exhibited the predicted pattern ( See Figure A.13): the reconstruction error for the slow changing feature was lower than the reconstruction error for the fast-changing feature.
> > > > >
> > > > > Why does the [no-memory] model outperforms the [multi-scale leaky memory + reset] model in test-error, in Section 5.2?
> > > > >
> > > > > The multi-scale model faces a challenge that was not faced by the single-scale models in Section 5.1: because all nodes in the hidden layer project to all nodes in the reconstruction later, the slowly changing hidden states of the model (the ones with longer memory) are contributing to the reconstruction of quickly-changing features in the data stream. There is a (small) cost in the overall test error, because slowly-changing internal states are ineffective for reconstructing quickly-changing features. We emphasize that the quantity of noise introduced is small, and that it is accompanied by a significant benefit in learning more interpretable, un-mixed representations of a multi-scale datastream. Nonetheless, this is an important trade-off, which we now demonstrate in the Appendix and in Figure A.14.
> > > > >
> > > > > To demonstrate  that “interference” from  slow units is indeed the source of poorer learning, we tested the hypothesis that (i) a higher correlation between hidden units with memory (units with short or long memory) and fast-changing part of the output would result in a worse performance in reconstructing the fast-changing feature; whereas (ii) a higher correlation between hidden unit with no-memory and fast-changing part of the output would not result in worse performance in reconstructing the fast-changing feature. These hypotheses were confirmed in our analyses shown in Figure A.14.

---

> > > > > > ### Author Response · Authors · 2020-11-25
> > > > > > **Our Response to Reviewer 2 (Part 6)**
> > > > > >
> > > > > > * Comment: The experiments are also in general not done very rigorously. For example, no hyperparameter tuning was done for the “smooth” case, but maybe the problem is just that the learning rate in this case should be slightly different (i.e. no need for special mechanisms like memory or gating). We can never know this unless the experiments are done more rigorously.
> > > > > >
> > > > > > We appreciate the reviewer’s concern about hyperparameters. We do not have the computational resources to exhaustively scan the space, but have consistently observed the primary empirical phenomenon (i.e.  the advantage of leaky memory with reset) across different datasets with different number of classes, different data dimensions, different amounts of category-overlap, and across different optimization and different cost functions. Also, to compensate for potential advantage of a specific set of hyperparameters for a specific condition, we ran several runs, each with a different random weight initialization, and reported the averaged results.
> > > > > >
> > > > > > We now report that the findings from Section 5.1 are reproduced for a range of learning rates. Also, we now report that for Section 5.2, we observed the same pattern of findings for a range of learning rates in the same multiscale dataset, as well as for a different synthesized multi-scale dataset (see Figure A.12).
> > > > > >
> > > > > > ------
> > > > > > Reference:
> > > > > > - Hadsell R. et al. “Embracing Change: Continual Learning in Deep Neural Networks”. Trends in Cognitive Sciences. Nov, 2020.
> > > > > > - Wu X. and Foster D. J. “Hippocampal Replay Captures the Unique Topological Structure of a Novel Environment”. Journal of Neuroscience. May, 2014.

---

### Official Review · AnonReviewer4 · 2020-10-28
**On the right track, but would like to see less artificial experiments and more realistic ones**

**Rating:** 6
**Confidence:** 4

**Review:**

Temporal smoothness is a recurring feature of real-world data that has been unaccounted for when training neural networks. Much of the random sampling in training neural networks is done to remove the temporal correlations originally present when the data is collected. This work aims to propose a method to train on this 'less processed' form of data.

There are two aspects of their method which makes training on smooth data possible:
1) Hidden units with 'multi-scale leaky memory'
2) Memory gating- between category transitions in time, memory is reset by setting $\alpha = 0$

For supervised learning, the authors create an artificially smooth dataset by presenting a model with examples from the same class repeatedly. They show as compared to a baseline model, their proposed method is able to learn effectively on highly repetitive data.

For unsupervised learning, the authors show that the model learns to match internal hidden unit representations with different \alpha = \{0.0, 0.3, 0.6\} with the corresponding timescale of \{fast, medium, slow\} features on a toy dataset.

Strengths:
- Specifies the right problem. Temporally smooth learning mechanisms are noticeably absent in the field.

- Interesting property where the proposed method does not use backpropagation through time despite having a recurrent hidden unit function

- Leaky memory is a simple idea to resolve this learning issue.

Places for improvement:

-Despite making the case that data in the real world is temporally smooth, the datasets which were used were artificially generated from a dataset that is not smooth (MNIST). Is there any issue in applying this method to a video segmentation as described in their example in Fig 1A?

-For the unsupervised learning scenario, the toy data only has variations that exactly match the timescales setup in the architecture of the network. What happens if these are mismatched with the input data?

-The accuracies also seem off for 1 repetition training of the baseline. The accuracy should be well above 90% on MNIST.

-How does mini-batching work in this temporally smooth data? Are the examples within a mini-batch temporally aligned? Temporally aligned mini-batches does not seem like a realistic assumption to make.

---

> ### Author Response · Authors · 2020-11-25
> **Our Response to Reviewer 4 (Part 1)**
>
> We thank the reviewer for the generally positive assessment and very helpful comments.
>
> * Comment: Despite making the case that data in the real world is temporally smooth, the datasets which were used were artificially generated from a dataset that is not smooth (MNIST). Is there any issue in applying this method to a video segmentation as described in their example in Fig 1A?
>
> We agree with the reviewer that the autocorrelation structure we are testing here does not match the autocorrelation structure of real-world visual information. This is an important next step for our work, which we are currently addressing by training our models on the “e-vds” dataset which shows short video clips of everyday objects, such as shoes, plants, and cups.
> That said, the problems of temporal autocorrelation in the setting of spatiotemporal-vision are quite task-specific: for example, one needs to employ a convolutional architecture and the temporal autocorrelation structure will reflect object motion as well as the relative motion of foreground and background features. It is unlikely that these forms of autocorrelation structure will generalize to other modalities (e.g. odor, audition), and certainly not to other levels of abstraction (e.g. learning of abstract semantic categories or situational schemas). Therefore, in our initial investigations, we have focused on a generic architecture and simplified stimulus sets in order to determine the basic principles, before applying these approaches to more particular cases with more immediate real-world applicability.
>
> Moreover, while we are working to extend to more realistic datasets, the toy datasets provide us with control necessary to determine some of the mechanisms and principles -- without toy data, we would not be able to perform manipulations like the ones shown in Appendix A.7, which helps to reveal the mechanism. Also, we are not only motivated to develop new machine learning approaches, but also seek to understand how learning unfolds in the human brain. From this perspective, the use of simplified datasets and generic architectures is not a drawback.
>
> Regarding the illustration in Figure 1.A, we apologize for the confusion. We meant to imply that data in the real world are correlated across nearby points in time, whereas data in training neural networks for categorization or reconstruction tasks are commonly randomized. In “smooth information in the real world” in Figure 1.A, different angles of the same face represent different samples of a category, identity of face structure of that person (i.e. each person being a different category). In “random sampling in training neural networks” in Figure 1.A, different faces represent shuffled samples from all categories. In the revised version of the paper, we modified Figure 1.A to make our point more clear and avoid confusion.

---

> > ### Author Response · Authors · 2020-11-25
> > **Our Response to Reviewer 4 (Part 2)**
> >
> > * Comment: For the unsupervised learning scenario, the toy data only has variations that exactly match the timescales setup in the architecture of the network. What happens if these are mismatched with the input data?
> >
> >
> > Thank you for this interesting and fruitful question. We have looked into this and have added new analyses to the revised version of the paper.
> >
> > (i) What does happen when the temporal structure of the data and the temporal properties of the model are different?
> >
> > To investigate this question, we trained our model on a data structure in which the consecutive items do not share any local features (no temporal correlation between consecutive items). In this setting, the leaky memory advantage was eliminated, and leaky memory was always less effective than learning without memory (See Appendix A.7). In this sense, we think of the leaky memory as a form of inductive bias, in which the model assumes that some task-relevant features are persistent over consecutive samples; when the data are consistent with this inductive bias, learning is accelerated.
> >
> > (ii) Can resetting be implemented in an unsupervised manner?
> >
> > In the original version of the paper, the resetting mechanism was implemented in a supervised way: the network was informed that “now is the time to reset the memory.”
> >
> > Inspired by bio-realistic resetting mechanisms, we now report that for learning multiscale data we have implemented a “resetting” mechanism in a straightforward and unsupervised way using only local computations, while preserving the same gains in learning efficiency. To do so, we have used the comparison between the difference and the average of the following input items as the resetting criterion, but other sorts of computations are also possible. Our implemented method is consistent with neurophysiological studies that demonstrate a sudden shift in memory representations in the face of a surprise in the input stimuli (DuBrow et al, 2017; Chien and Honey, 2020).
> >
> > The more bio-plausible implementation of event-related resetting was implemented as follows: We reset the memory when the difference between the consecutive inputs is larger than their average. For instance, the memory of the hidden node with long memory will be reset based on the amount of change in the slow-changing feature of the input.
> >
> > [ t represents the iteration number during training, It is the current state, It-1 previous state ]
> > | I(t) - I(t-1)| >  |I(t) + I(t-1)| / 2
> >
> > (iii) What happens if the memory timescales do not match the reset timescales?
> >
> > For learning multiscale data, in the original paper, for implementing memory and reset, we had used a [multiscale leaky memory + reset] model (alpha being 0, 0.3, and 0.6). In that case, the length of the memory of hidden nodes matched their resetting timescale: e.g. long-memory node was reset based on change in slow-changing feature.
> >
> > We now report that we have tested an extra condition in which all hidden units have the same scale of leaky memory (alpha being 0.5, 0.5, and 0.5), but each node has a different resetting mechanism: one node is sensitive to the change in the fast-changing feature, another node is sensitive to the change in medium-changing feature, and the last node is sensitive to the change in the slow-changing feature. This model is called [leaky memory + reset].
> >
> > We found that [multiscale leaky memory + reset] and [leaky memory + reset] performed similarly in efficiency and timescale-selectivity criteria (See Figure 3). Therefore, when learning multi-timescale data, even if the memory does not have multiple timescales but the resetting mechanism is sensitive to different timescales, the model can learn the different temporal properties of the information.
> >
> > Regarding the resetting mechanism using the information from input data, this resetting method is consistent with neurophysiological studies that demonstrate a sudden shift in memory representations in the face of a surprise in the input stimuli (DuBrow et al, 2017; Chien and Honey, 2020). Implementing a resetting mechanism that does not use information from input is an interesting future step of this project. We are currently working on implementing resetting mechanisms that use local prediction error to identify the timing of reset.

---

> > > ### Author Response · Authors · 2020-11-25
> > > **Our Response to Reviewer 4 (Part 3)**
> > >
> > > * Comment: The accuracies also seem off for 1 repetition training of the baseline. The accuracy should be well above 90% on MNIST.
> > >
> > > Thanks for pointing this out. In Figure 2, to visualize the difference across different conditions, we are showing the results early on in the training (first 4000 iterations of the first epoch). The accuracy after multiple training epochs is much higher than what shown in Figure 2. Also, we should mention that the implemented model is a simple 3-layer neural network without any convolutional structure.
> > >
> > >
> > > * Comment: How does mini-batching work in this temporally smooth data? Are the examples within a mini-batch temporally aligned? Temporally aligned mini-batches does not seem like a realistic assumption to make.
> > >
> > > Thanks for pointing this out. We have performed new analyses and now report that in our revised paper we have taken a closer look into the interaction between mini-batching and temporal smoothness.
> > >
> > > To investigate the interaction between temporal correlation and mini-batching, we have explored the following questions:
> > >
> > > (ii) How do batching and smoothness interact, when they are temporally aligned?
> > >
> > > We have performed new analyses in which batch size and smoothness are temporally aligned (See Figure A.5, mini-batching).
> > > Regarding the assumption of alignment between the smoothness and the batch size, the original motivation for this analysis was to compare the effects of averaging in mini-batching and averaging in leaky memory (The comparison shown in Figure A.5). However, we also looked at situations where the batch size and the smoothness do not match, reported below.
> > >
> > > (ii) How do different amounts of temporal correlation affect batch-training for a specific batch size?
> > >
> > > In Appendix A.6, we show the results of different levels of temporal smoothness for a batch of size 16. Our results showed that  smoothness  does  not  influence  mini-batch  training  similarly  to  SGD  training.   Early  in  the training, minimum smoothness showed the fastest learning and higher levels of smoothness showed slower learning. However, later in the training, another pattern was observed: the condition with the smoothness level equal to the batch size (e.g. 16 repetitions for batch of 16) showed the greatest learning efficiency compared to both lower levels of smoothness (e.g. 10 repetitions) and higher levels of smoothness (e.g. 24 repetitions).
> > >
> > > In connecting the mini-batch data to the results reported in Figure (2), consider that ”smoothness” can happen at 2 levels:  samples can be similar to one another within a batch (”smooth within a batch”) and the composition of samples can be similar across consecutive mini-batches (”smooth across batches”).  It seems that early in the training,  the conditions with minimum ”within-batch smoothness” have the highest learning speed;  this makes sense as the composition of each mini-batch is most reflective of the overall composition of the test data.  However, later in the training, the condition with minimum ”across-batch smoothness” has the best learning speed.  Minimum across-batch smoothness refers to the condition where each batch consists of items from only one category (e.g. 16 repetitions for batch of 16).  Note that when each individual batch contains items all from one category, this also implies that consecutive batches will not contain any items from the same categories, leading to a ”minimum smoothness at the batch level”. Thus, the advantage for minimum across-batch smoothness may be analogous to what is observed in Figure 2 at the single item level, but occurring at the batch level.
> > >
> > > Future work is required to further investigate how smooth data interact with mini-batch training.
> > >
> > > --------
> > >
> > > Reference:
> > >
> > > - Sarah DuBrow, Nina Rouhani, Yael Niv, and Kenneth A Norman. Does mental context drift or shift? Current opinion in behavioral sciences, 17:141–146, 2017.
> > >
> > > - Hsiang-Yun Sherry Chien and Christopher J Honey. Constructing and forgetting temporal context in the human cerebral cortex. Neuron, 2020.

---

### Official Review · AnonReviewer1 · 2020-10-28
**Authors address a very relevant and interesting question**

**Rating:** 6
**Confidence:** 4

**Review:**

### Update after response:

The authors have quite thoroughly addressed most of my concerns with updates and new experiments. So I will increase my score to an accept at this point.

Summary:

The authors consider the question of the effect of temporally correlated data on learning. They show that while standard networks are adversely affected by this smoothness, using mechanisms such as leaky memory in activation units and memory gating allows networks to take advantage of this data. The authors also further study the representations that emerge from said mechanisms.

Overall, the work is quite interesting and insightful. But some of the concerns listed below put this work below the threshold of acceptance for me.

Strengths:

+ The question is very well motivated and relevant, since, as the authors point out, a lot of data in the real world is highly temporally correlated.
+ This work provides a nice compact explanation for the role of leaky processes in biology.

Weaknesses:

- With temporally correlated data, once expects that the ability of the network to temporally process the data would also have a significant effect on performance. In this context, it seems to me that a comparison with RNNs is also quite relevant as a baseline.
- The Auto-encoder with no memory seems to perform better than other models. This needs an explanation at the least. This could either indicate that the experiment they consider is too simple, or it could point to some more fundamental underlying issues with the combination of auto-encoders and sequentially correlated data. Depending on which, the rest of the analysis may not generalise. This is a major weakness in the paper.
- A more detailed explanation for why BPTT is not used (or comparison of all results with BPTT baselines) would make the results a lot more useful. BPTT implies propagating the gradient through the leaky memory.
- A discussion of the universal assumption of i.i.d data in machine learning deserves more space in the introduction.

Other questions:

* Could the authors comment on what sort of processes could control event-related resetting in biology?

Minor:

* Defining what incremental learning is early on would improve the clarity of the paper.
* Visualisation of the dynamics of units of the AE could prove to be quite interesting.
* The first line of section 4.1.2 is a bit confusing: "We tested MNIST, Fashion-MNIST, and further synthetic datasets containing low category overlap" seems to imply MNIST and Fashion-MNIST have low category overlap, which is not the case and I don't think that's what the authors meant to imply.

---

> ### Author Response · Authors · 2020-11-25
> **Our Response to Reviewer 1 (Part 1)**
>
> We thank the reviewer for the generally positive assessment and appreciation of our work, and for very helpful comments.
> * Comment: With temporally correlated data, one expects that the ability of the network to temporally process the data would also have a significant effect on performance. In this context, it seems to me that a comparison with RNNs is also quite relevant as a baseline.
>
> We thank the reviewer for pointing this out. In the original version of the paper, we did not implement LSTM since it uses backpropagation-through-time (BPTT), which is implausible for biological settings. In learning with BPTT, the same neurons must store and retrieve their entire activation history [Sutskever, 2013; Lillicrap & Santoro, 2019]. The learning in [leaky memory + reset] model is more biologically plausible and computationally simpler because it does not require maintaining the whole history or computing a gradient relative to that history.
>
> Nonetheless, we agree with the reviewer that RNNs provides a useful baseline comparison, as they indicate how efficient the learning is, relative to a learning system that is optimized for combining information over time. Therefore we implemented an LSTM model to classify MNIST dataset and we compared the performance of the LSTM model against the [leaky memory + reset] models.
> The LSTM model characteristics are: 1 layer of LSTM (input size= 784, hidden size = 392) followed by 1 linear layer of size (392, 10). The loss function, optimization method, and learning rate were identical to the ones used for the [leaky memory + reset] models.
> Overall, as indicated in the new analyses below, we found that the LSTM model learned categories slightly better than any of the models we examined, but it’s performance did not generalize across timescales.
> To arrive at these findings, we : (i) investigated the effects of levels of smoothness in training data on category-learning in LSTM; and (ii) compared the memory-reset model with LSTM under similar conditions.
>
> (i) Effects of smoothness on LSTM
>
> We found that LSTM was affected by smoothness in a similar way to the memory-reset model, i.e. higher smoothness in training data results in better classification performance (See Appendix A.8).
>
> (ii) Generalization of LSTM and leaky-memory models to variations in temporal structure
>
> Both the LSTM and the [leaky memory + reset] model have a mechanism for  forgetting information. However, the LSTM has a much more flexible architecture which enables it to alter how much information it preserves about prior stimuli, calibrating its memory to the exact structure of the training data. We hypothesized that the LSTM would therefore have a particular advantage when being tested on data with the same temporal structure that it has been trained on, but it may not generalize as well to data in which the temporal structure is altered. To investigate this possibility we performed the following analysis:
> We first tested both the LSTM and memory-reset on the same data structure that they were trained on (e.g. training with 5-repetitions and testing on 5-repetitions). This means that we included memory in the testing process and tested the models on the same order that they were trained on. In this setting, the LSTM was the best model, and the [leaky memory + reset] model was second-best (See Appendix A.9).
>
> We then tested both the LSTM and [leaky memory + reset] models on a different structure from which they were trained on (e.g. training with 5-repetitions and testing on 1-repetition). Consistent with our hypothesis, in this setting, the [leaky memory + reset] model outperformed the LSTM (See Appendix A.10). Differences between the LSTM and the [leaky memory + reset] model suggest that the two do not exploit the temporal structure of the data stream in the same way. The LSTM makes use of information about the specific task structure (e.g. there are precisely 5 repetitions in a block) and its performance is reduced when this assumption is violated in generalization data. Conversely, the [leaky memory + reset] model simply uses the temporal smoothness in the training data to learn more useful internal representations. These representations are still useful even when the model must categorize items individually, and is unable to make use of any temporal smoothness at test.

---

> > ### Author Response · Authors · 2020-11-25
> > **Our Response to Reviewer 1 (Part 2)**
> >
> > * Comment: The Auto-encoder with no memory seems to perform better than other models. This needs an explanation at the least. This could either indicate that the experiment they consider is too simple, or it could point to some more fundamental underlying issues with the combination of auto-encoders and sequentially correlated data. Depending on which, the rest of the analysis may not generalise. This is a major weakness in the paper.
> >
> > Thanks for pointing this out. We have looked into this more closely, and we believe that this was helpful for better understanding how these multi-scale systems operate. Overall, our analyses indicate that the findings from Section 5.1 and Section 5.2 are complementary rather than contradictory.
> >
> > (i) Does faster convergence in the no-memory model from Section 5.2 contradict the benefit of the memory-reset model from Section 5.1?
> >
> > We do not believe that the results are contradictory. Consider that the multiscale data stream in Section 5.2 is composed of three different subcomponents (top, middle and bottom rows of the input). Thus, the multi-scale stream can be understood  as a combination of the 1-rep condition from Section 5.1 (where the feature changes with every sample), the 3-rep condition from Section 5.1 (where the feature changes at a medium speed across samples), and the 5-rep condition from Section 5.1 (where the the feature changes slowly across samples).
> >
> > In Section 5.1, we showed that the [leaky memory + reset] model performs better when the data has higher smoothness (e.g. in Figure 2.C, category learning is more efficient for 5-repetitions than for 1-repetition). If the same pattern holds in the context of multi-scale autoencoders, for models with memory and reset, we should see that the slow-changing feature (5-repetitions) is more quickly learned than the fast-changing feature (1-repetition). To test this, we measured the per feature error (i.e. test error for reconstructing a specific feature) and we compared the fast and slow-changing features. Consistent with the pattern observed in section 5.1 of the paper, we saw that both the [leaky memory + reset] model and [multiscale leaky memory + reset] model exhibited the predicted pattern ( See Figure A.13): the reconstruction error for the slow changing feature was lower than the reconstruction error for the fast-changing feature.
> >
> >
> > (ii) Why does the [no-memory] model outperforms the [multiscale leaky memory + reset] model in test-error, in Section 5.2?
> >
> > The multi-scale model faces a challenge that was not faced by the single-scale models in Section 5.1: because all nodes in the hidden layer project to all nodes in the reconstruction later, the slowly changing hidden states of the model (the ones with longer memory) are contributing to the reconstruction of quickly-changing features in the data stream. There is a (small) cost in the overall test error, because slowly-changing internal states are ineffective for reconstructing quickly-changing features. We emphasize that the quantity of noise introduced is small, and that it is accompanied by a significant benefit in learning more interpretable, un-mixed representations of a multi-scale datastream. Nonetheless, this is an important trade-off, which we now demonstrate in the Appendix and in Figure A.14.
> >
> > To demonstrate  that “interference” from  slow units is indeed the source of poorer learning, we tested the hypothesis that (i) a higher correlation between hidden units with memory (units with short or long memory) and fast-changing part of the output would result in a worse performance in reconstructing the fast-changing feature; whereas (ii) a higher correlation between hidden unit with no-memory and fast-changing part of the output would not result in worse performance in reconstructing the fast-changing feature. These hypotheses were confirmed in our analyses shown in Figure A.14.
> >
> > Regarding the generalizability of multiscale autoencoder models, we now report that we found the same pattern for a range of learning rates, as well as a different multiscale synthesized dataset (See Appendix A.12). Also, we have run 50 runs each with a different random weight initialization to prevent a specific learning-rate interacting with the weight-initialization of a particular model run. We reported the average results across all 50 runs.

---

> > > ### Author Response · Authors · 2020-11-25
> > > **Our Response to Reviewer 1 (Part 3)**
> > >
> > > * Comment: A more detailed explanation for why BPTT is not used (or comparison of all results with BPTT baselines) would make the results a lot more useful. BPTT implies propagating the gradient through the leaky memory.
> > >
> > > Thanks for the constructive feedback.
> > >
> > > As reported in response to point (1) regarding comparison with RNN, we now have an LSTM model which implements BPTT, and we compare the LSTM model to the [leaky memory + reset] model. These new analyses can be found in Appendix A.8, A.9, and A.10.
> > >
> > > In the revised version of the paper, we make it clear in the Introduction that one of our objectives in testing these leaky memory architectures is to perform learning without requiring gradients to flow into prior timepoints, because we seek simple and biologically plausible approaches.
> > >
> > >
> > > * Comment: A discussion of the universal assumption of i.i.d data in machine learning deserves more space in the introduction.
> > > We thank the reviewer for pointing this out.
> > >
> > > Thanks for pointing this out. We now report that we added a brief discussion about the assumption of i.i.d data in machine learning to section “Related Work” under “Potential costs and benefits of training with smooth data”.
> > >
> > >
> > > * Comment: Could the authors comment on what sort of processes could control event-related resetting in biology?
> > >
> > > This is an interesting question.
> > >
> > > Despite continuous sensory input, our subjective experience is discrete, perceiving events with identifiable beginnings and ends (Radvansky & Zacks, 2018). There are different theories about the perception of event-boundaries in the brain.
> > >
> > > One possibility is that processings in the brain can detect boundaries in the input signal by calculating “prediction error” which is the difference between the expectation (prediction) and observation. When prediction errors (PE) arise, an event boundary is perceived. For instance, in an fMRI study, Zack et al (2011) found that Basal Ganglia -- a brain region known for signaling PE -- showed increased neural response across event-boundaries compared to within event-boundaries. The ability of computing PE is not unique to Basal Ganglia. Neurophysiological data show that other brain regions, such as Anterior Cingulate Cortex and Subcortical Neuromodulatory systems, can also compute PE and implement it for event perception (Zack et al, 2011). These findings support that signaling PE in the brain and perceptions of event-boundaries are correlated.
> > >
> > > Another piece of evidence from the brian’s perception of event-boundaries comes from studies on Hippocampus, one of the main brain regions for learning and memory. It has been shown that when there is an abrupt change in the stimuli, hippocampal activities shift suddenly and create discontinuity in the memory (DuBrow et al, 2017).
> > >
> > > Inspired by bio-realistic resetting mechanisms, we now report that for learning multiscale data we have implemented a “resetting” mechanism in a straightforward and unsupervised way using only local computations, while preserving the same gains in learning efficiency. To do so, we have used the comparison between the difference and the average of the following input items as the resetting criterion, but other sorts of computations are also possible. Our implemented method is consistent with neurophysiological studies that demonstrate a sudden shift in memory representations in the face of a surprise in the input stimuli (DuBrow et al, 2017; Chien and Honey, 2020).
> > >
> > > The more bio-plausible implementation of event-related resetting was implemented as follows: We reset the memory when the difference between the consecutive inputs is larger than their average. For instance, the memory of the hidden node with long memory will be reset based on the amount of change in the slow-changing feature of the input.
> > >
> > > [ t represents the iteration number during training, It is the current state, It-1 previous state ]
> > >  |I(t) - I(t-1)|  > |( I(t) + I(t-1) ) / 2|

---

> > > > ### Author Response · Authors · 2020-11-25
> > > > **Our Response to Reviewer 1 (Part 4)**
> > > >
> > > >
> > > > *  Minor comment: Defining what incremental learning is early on would improve the clarity of the paper.
> > > >
> > > > Thanks for pointing this out. We now report that we have added a brief description about incremental learning to section “Introduction” to further clarify our motivation.
> > > >
> > > > * Minor comment: Visualisation of the dynamics of units of the AE could prove to be quite interesting.
> > > >
> > > > Thanks for your interest. We looked into this and visualized dynamics of the hidden units of different models shown in Appendix A.15. Figure A.15 shows how hidden units in different models extract the timescale of the input stream.
> > > >
> > > >
> > > > * Minor comment: The first line of section 4.1.2 is a bit confusing: "We tested MNIST, Fashion-MNIST, and further synthetic datasets containing low category overlap" seems to imply MNIST and Fashion-MNIST have low category overlap, which is not the case and I don't think that's what the authors meant to imply.
> > > >
> > > > Thank you for pointing this out, and we apologize for the confusion. We meant: "We tested MNIST, Fashion-MNIST, and synthetic datasets containing low category overlap." (updated in the revised paper)
> > > >
> > > > -----------------
> > > > Reference:
> > > >
> > > > - Zacks JM, Speer NK, Swallow KM, Braver TS, Reynolds JR. Event perception: a mind-brain perspective. Psychol Bull. 2007 Mar;133(2):273-93.
> > > >
> > > > - Richmond, L. L., Gold, D. A., & Zacks, J. M. (2017). Event perception: Translations and applications. Journal of applied research in memory and cognition, 6(2), 111–120.
> > > >
> > > > - Sarah DuBrow, Nina Rouhani, Yael Niv, and Kenneth A Norman. Does mental context drift or shift? Current opinion in behavioral sciences, 17:141–146, 2017.
> > > >
> > > > - Hsiang-Yun Sherry Chien and Christopher J Honey. Constructing and forgetting temporal context in the human cerebral cortex. Neuron, 2020.

---

### Author Response · Authors · 2020-11-24
**Our general response to all reviewers**

We thank all reviewers for carefully reading our manuscript, and providing valuable and constructive suggestions. We have revised the manuscript thoroughly according to the comments.

In particular, the main revisions that we made are as follows:

1) We have compared the [memory + reset] model to an LSTM model trained with backprop-through-time (BPTT), and have also tested the generalization of both classes of models when they are trained on one temporal structure and tested on a different temporal structure;

2) We have provided a new demonstration of a case in which leaky-memory mechanisms are actually  disadvantageous, and we use this to provide a more mechanistic explanation (Section 5.1.3) for why leaky memory and resetting mechanisms improve learning from temporally correlated data;

3) We have more deeply analyzed how the activation-averaging in the [memory + reset] model compares with gradient-averaging in mini-batching approaches;

4) We replicated the findings in Figure 3 across a range of learning rates, and different kinds of synthetic multiscale data;

5) We added a new paragraph to Section 5.2.3 discussing why the no-memory AE model produces slightly lower reconstruction error than feedforward models;

6) We clarified the Introduction section to better motivate our work, explaining why the incremental-learning settings important and why we pursue the goal of learning without BPTT;

In addition, we made the following minor revisions:

7) We modified Figure 1 to more clearly reflect the kinds of analyses that we ran;
8) We modified Figure 2 so that the random sampling in all plots is identical and can be used as a common reference;
9) We combined the original Figure 3 and Figure 4 into a new Figure 3;
10) We have extended the section on Related Work;
11) We added 9 new Appendix sections, providing the figures to support the new analyses.;

Overall, we are very thankful for all the suggestions that helped us to clarify our motivation, sharpen our questions, broaden our findings, and strengthen the science.

---

### Decision · Program_Chairs · 2021-01-07
**Final Decision**

**Decision:**

Reject

**Comment:**

It appears that this paper can benefit from additional detail and work before it becomes a stronger publication that is more convincing. The authors have done an impressive job responding to the reviewers and updating their paper, and multiple reviewers raised their score consequently. However, while multiple reviewers now recommend acceptance, there is no agreement on it. Even among the reviewers who recommended acceptance, there is a feeling on being on the fence specifically about the ability of the paper to make a convincing argument without considering a real life scenario and while only using toy settings. Indeed, this is a problematic aspect of the paper because the value of the paper lies in making that argument. Further, the paper would gain further from clarifying the writing further and connecting the paper more directly with the neuroscientific literature it aims to be connected to.